# The effect of overnight consolidation in the perceptual learning of non-native tonal contrasts

Zhen Qin[1,2]*, Caicai Zhang[1,2]*

**1** Department of Chinese and Bilingual Studies, The Hong Kong Polytechnic University, Hong Kong SAR, China, **2** Research Center for Language, Cognition, and Neuroscience, The Hong Kong Polytechnic University, Hong Kong SAR, China

* caicai.zhang@polyu.edu.hk (CCZ); zhen-quentin.qin@polyu.edu.hk (ZQ)

**Data Availability Statement:** All data files are available from the OSF platform: https://osf.io/284j9/.

**Funding:** This work was supported in part by the Departmental General Research Funds

## Abstract

Sleep-mediated overnight consolidation has been found to facilitate perceptual learning by promoting learners' generalization across talkers in their perception of novel segmental categories. Lexical tone is characterized by high variability across talkers, and displays dynamic change over time. For this reason, it remains unclear whether a similar effect of overnight consolidation would be found for perceptual learning of novel tonal contrasts. Thus, this study aims to examine whether overnight consolidation facilitates talker-independent learning of lexical tones in the identification and discrimination of novel Cantonese level tones by Mandarin listeners. Two groups of Mandarin listeners were perceptually trained either in the morning or in the evening. Listeners were trained in a tone identification (ID) task with feedback using stimuli produced by a trained talker. Their post-training changes and generalization to a novel talker were then tested in the ID and AX discrimination tasks using stimuli produced by trained and untrained talkers in three posttests following training: immediately after training, 12-hour delay, and 24-hour delay. While the evening group slept between the first and second posttests, the morning group did not. The accuracy rates in the ID task showed that the evening group showed an improved trend, predicted by their individual sleep time, in identifying the level tones produced by both the trained and untrained talkers; in contrast, the morning group showed a declining trend. The d-prime scores in the AX discrimination task did not show different patterns between the two groups. The finding of sleep-related identification changes over time suggests that overnight consolidation might have facilitated tone learning of stimuli produced by the novel talker and eventually facilitated the formation of a more talker-independent representation of novel tone categories in long-term memory. The results are discussed in light of the features of lexical tones to shed light on the mechanism of phonetic learning.

## 1. Introduction

Converging evidence indicates that sleep supports various aspects of language learning by facilitating the memory consolidation of newly learned knowledge [1,2]. Sleep-mediated

(International collaboration) and the Departmental Reward Scheme for Research Publications in Indexed Journals awarded to CCZ, and the Language Learning Early Career Research Grant and the Postdoctoral Fellowships Scheme at the Department of Chinese and Bilingual Studies of the Hong Kong Polytechnic University awarded to ZQ. The funders had no role in study design, data collection and analysis, decision to publish, or preparation of the manuscript.

**Competing interests:** The authors have declared that no competing interests exist.

memory consolidation (i.e., overnight consolidation) plays an important role in novel word learning [3–7]. For instance, one study found that only the set of novel words learned in the evening and consolidated during the overnight interval, compared with another set of novel words learned during the daytime without overnight consolidation, showed a lexical competition with existing words and elicited faster responses than those learned during the daytime [6]. In another study, participants showed an immediate advantage for trained affixes in a speeded shadowing task if these affixes occurred in the stem contexts in which they were learnt (e.g., buildnule), and this learning effect was generalized to words with untrained stems (e.g., sailnule) only after the trained words were consolidated after an overnight sleep [8]. The above findings supported a two-stage complementary learning systems (CLS) model, which postulates that (word) learning consists of an initial episodic encoding of context-dependent information, followed by a longer-term, sleep-mediated, consolidation process (e.g., through declarative memory system) in which context-independent lexical representations emerge [9,10].

On the other hand, a growing literature showed that overnight consolidation also facilitates listeners' perceptual learning of speech sound categories in an auditory domain [11–15]. It is interesting that similar findings were reported, considering that learners' sound representations could differ from their word knowledge in multiple aspects. For instance, previous work [11] suggests that a period of sleep following perceptual training of speech is associated with generalization of speech information to unfamiliar word contexts in mapping synthetic tokens to native categories. More specifically, an immediate generalization in speech sound learning declined over the course of a day, and then rebounded after a night of sleep. This suggests that the memory of the newly-induced (context-independent) abstraction might degrade during waking periods but be stabilized during sleep, which might have facilitated the recovery and subsequent retention of context-independent sound learning [11,12]. However, the studies above have all addressed how sleep affects perceptual adjustments within one's native language; thus, how sleep assists the acquisition of non-native sounds remains under-studied.

Non-native speech sounds are perceptually difficult for adults to learn, particularly when the sounds are similar, but not identical, to the sounds of their native language (L1) [16,17]. However, the conditions under which the learning of new sounds is facilitated or inhibited are unclear. In recent studies, Earle and her colleagues examined how learned sound categories were encoded into long-term memory following a session of laboratory training by focusing on the effect of overnight consolidation in the identification and discrimination of novel phonetic categories, that is, a Hindi dental and retroflex stop contrast [18–22]. Specifically, Earle and Myers showed that overnight consolidation promoted English listeners' generalization across talkers in their identification, but not in the discrimination, of the novel Hindi contrast [19]. A pretest-training-posttest paradigm was conducted on two groups of English listeners who were perceptually trained either in the morning or in the evening, following this procedure: (i) listeners were perceptually trained using stimuli produced by a trained talker in an identification task; (ii) listeners' post-training perceptual changes were tested in an identification task and an AX (category) discrimination task (with a long ISI, that is, 1000ms) using stimuli produced by trained and untrained talkers. Importantly, the identification task, relying on explicit recall of sound categories, and the category discrimination task, using a listener's implicit ability to automatically direct attention toward the acoustic cues in the signal, were argued to reflect different aspects of listeners' memory system, and thus were both included in posttest sessions [15,19]. Listeners' post-training perceptual changes were assessed in series of posttests at three time points over 24h following training: immediately after training, 12-hour delay, and 24-hour delay. Whereas the evening group slept between the first and second posttests, the morning group did not. The results of identification tests showed that while English

listeners trained in the evening improved significantly in identifying the stop stimuli produced by the untrained talker, those trained in the morning did not show such a pattern. In contrast, the two training groups did not show perceptual changes in identifying the stop stimuli produced by the trained talker. The results of the AX discrimination tests did not show a difference of the two training groups in their post-training perceptual changes. The discrepancy between the identification and discrimination tasks was attributed to the greater variability (untrained vowel contexts) of the discrimination task, which may have made it difficult for listeners to use consistent criteria in performing the task. However, an alternative explanation is that the underlying mechanisms may differ between the identification and discrimination tasks, which might have tapped into listeners' knowledge at different levels. Consistent with the CLS account, the finding of the identification task indicates that overnight consolidation should have facilitated the abstraction of novel sound categories, that is, a transfer of episodic sound information from an acoustic-sensory-based trace to a more talker-independent representation of the target stop contrast in long-term (e.g., declarative) memory, which explains the generalization to the untrained talker.

Moreover, a similar effect of overnight consolidation on the abstraction of novel phonetic information was found in lexically-guided phonetic retuning of non-native segments, in that overnight consolidation facilitated generalization of accent adaptation to a new (i.e., untrained) Mandarin talker by helping English listeners abstract away from specific properties of the trained talker in lexically-guided phonetic retuning [23].

The overnight consolidation effect in talker generalization (i.e., talker-independent abstraction) found for the perceptual learning and lexically-guided phonetic retuning of non-native segments, albeit informative, raises the question of whether a similar effect would be found in the perceptual learning of non-native tones. Compared to segments (e.g., Hindi dental and retroflex stops), which have been the focus of investigation in previous studies [19,22], lexical tones are highly variable and dynamic, requiring listeners to evaluate the pitch of the signal against the pitch range of the talker and continuously update this evaluation as the pitch contour unfolds over time [24–28] (see Fig 1 for the pitch contour of Cantonese and Mandarin tones). Further complicating the perceptual process, the same talker also shows variability in the production of tonal categories, thus also requiring listeners to evaluate the pitch they hear against the talker's different realizations of the same tonal categories (within-category variations) and of different tonal categories (cross-category variations) [29–35]. Thus, it is both important and challenging for listeners to extract the abstract representations from tonal exemplars in order to generalize successfully across talkers [30]. The variable and dynamic features of lexical tones were reported to induce a higher degree of sensitivity to mispronunciations of tones (i.e., a change in tone identity) than those of vowels and consonants in spoken word recognition for children and adult listeners [36–38]. For this reason, it is critical to verify whether a similar sleep-mediated talker-independent learning of novel tonal contrasts as that reported on the learning of novel segmental contrasts can be found, and to probe the time-course of post-training perceptual changes in tone learning.

Suprasegmental information (e.g., pitch) is important for distinguishing among words in Chinese languages. For instance, as illustrated in Fig 1, segmentally identical words that contain different lexical tones in Cantonese (/si 55/ 'silk' (Tone 1 (T1)), /si 25/ 'history' (T2), /si 33/ 'to try' (T3), /si 21/ 'time' (T4), /si 23/ 'city' (T5), and /si 22/ 'matter' (T6) [39]) and in Mandarin (e.g., /paā/ 'eight' (T1), /paá/ 'to pull out' (T2), /pa˜/ 'to hold' (T3), /paà/ 'father' (T4) [40]) differ in meaning. While pitch is the primary cue in tone perception, listeners with different language backgrounds attend to different pitch dimensions under the influence of their L1 prosodic system [41–44, also see 45,46]. Cantonese listeners were found to use both pitch contour (i.e., tone shape; falling vs. rising tones) and pitch height (i.e., average height; higher vs.

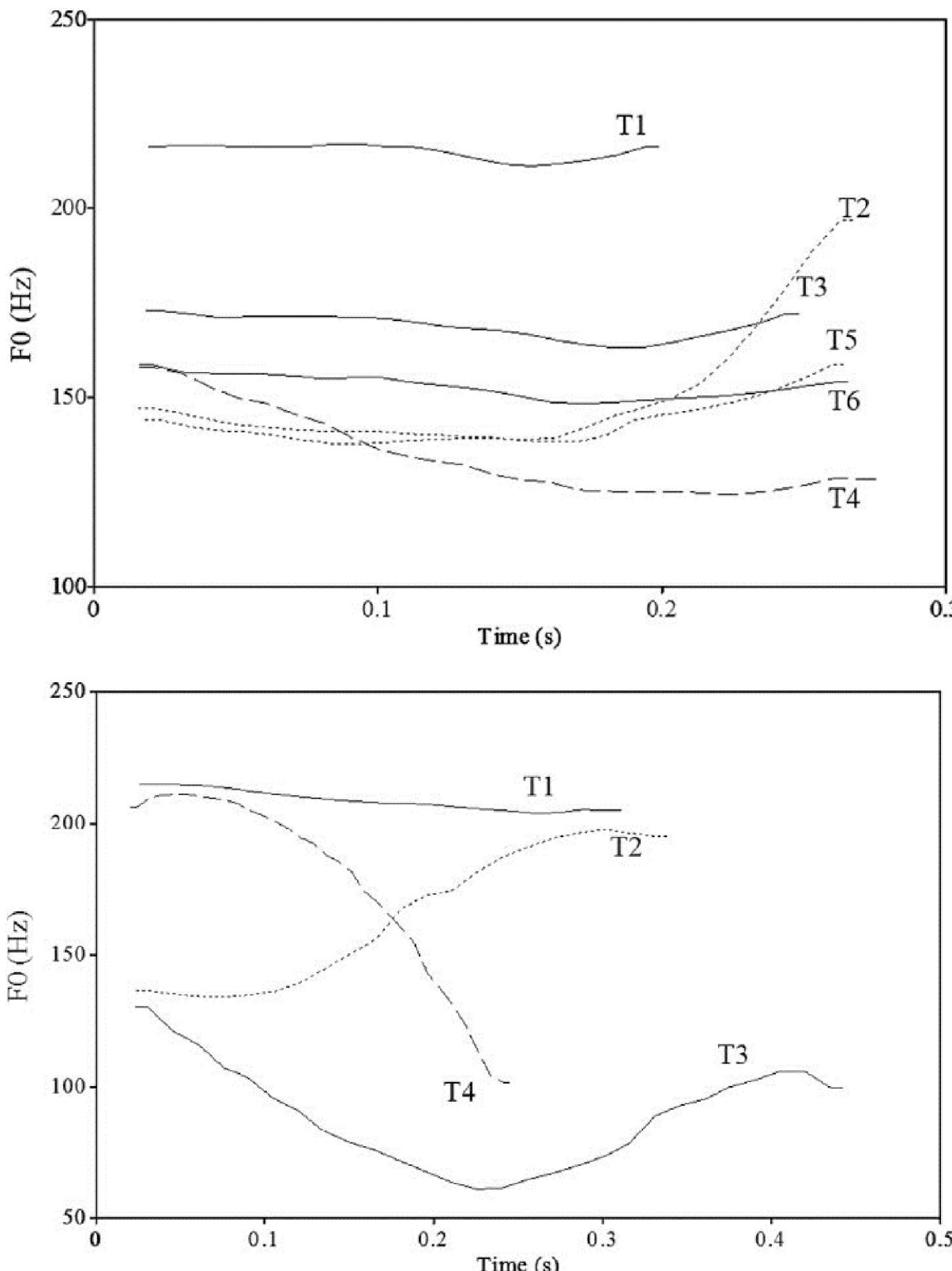

**Fig 1. Tone contours of the six Cantonese tones and four Mandarin tones.** The Cantonese tones (top panel) were produced by a female native speaker of Hong Kong Cantonese. The Mandarin tones (bottom panel) were produced by a male native speaker of Beijing Mandarin. Both figures were adopted from Qin & Jongman, 2016 for the purpose of illustration.

lower tones) to distinguish their native tonal contrasts [41,44]. Some Cantonese tone pairs, for instance, T3 (mid-level)–T6 (low-level), are distinguished in subtle differences in pitch height, and thus pose a great perceptual difficulty for listeners [47–49]. In contrast, Mandarin listeners are more sensitive to pitch contour than pitch height [50–53], and have been reported to have a great difficulty distinguishing (e.g., Cantonese) level-level tonal contrasts [41,54–57]. This difficulty may be explained by several reasons: first, a level-level tonal contrast is a novel contrast for Mandarin listeners given that no such contrasts exist in Mandarin tone inventory (see Fig 1); second, Mandarin listeners show reduced sensitivity to pitch height differences, compared with non-tone language listeners, due to their tonal categorization contrasting in pitch contour [58,59].

Given the Mandarin listeners' perceptual difficulty of level-level tonal contrasts and the variable/dynamic nature of level tones [29,60], the current study aims to examine whether a sleep-mediated talker-independent learning of novel tones, implying the abstraction of novel tonal categories, can be found in the perceptual learning of Cantonese level-level tonal contrasts (T1 /55/, T3 /33/, and T6 /22/) by Mandarin listeners. In light of previous findings on the learning of novel segmental contrasts, it is hypothesized that a similar effect of overnight consolidation will be found for the perceptual learning of Cantonese level tones by Mandarin listeners. Specifically, those listeners who are trained in the evening are expected to perform better than those who are trained in the morning in perceiving the level tones produced by a new talker and probably also by the trained talker, given a potentially greater difficulty of learning the target (three-way) tonal contrasts relative to (two-way) segmental contrasts [61–63]. As discussed above, the magnitude of the overnight consolidation effect and underlying mechanisms may differ between the identification and discrimination tasks [15,19]. For this reason, we included both identification and discrimination tasks to test whether the overnight consolidation effects found for the identification can be transferred or generalized to the discrimination of tonal contrasts by controlling stimuli variability in both tasks. It is hypothesized that a similar effect will be found in both the identification and discrimination tasks if different degrees of stimuli variability accounted for the discrepancy of the two tasks in terms of the overnight consolidation effect in the previous study [19]. Finally, it is expected that the overnight consolidation effect would be across-the-board for the perceptual learning of all three level tones.

## 2. Materials and methods

### 2.1 Participants

Thirty-three students (19 female, 14 male) between the ages of 18 and 30 were recruited using an online self-report questionnaire from the Hong Kong Polytechnic University (PolyU). Mandarin-speaking participants who had a minimal exposure to Cantonese were selected based on the following criteria: (1) having resided in Hong Kong for less than ten months and learned Cantonese in classroom less than one month prior to the pretest session, (2) speaking Mandarin as their mother tongue and not any Southern dialect including Shanghainese, Hakka, and Southern Min, and (3) having not received professional musical training. None of them reported history of hearing impairment, neurological illness, and sleep disorder. One participant was excluded on the basis of a late reporting of exposure to musical (i.e., piano) instruction. Although no power analysis was performed for the calculation of sample size, the sample size of the morning and evening groups in the current study was largely comparable to that reported in previous studies [19,23].

The remaining 32 participants were equally and randomly assigned into one of the two groups who were trained either in the morning (8–10 am) or evening (8–10 pm). Demographic characteristics of the morning and evening groups are summarized in Table 1. A set of

**Table 1. Demographic characteristics of the morning and evening training groups.**

| | Morning | Evening | *p*-value |
|---|---|---|---|
| No. of participants | 16 (9 F, 7 M) | 16 (9 F, 7 M) | / |
| Age (year) | 24.9 (3.7) | 23.3 (3.5) | *p* = .23 |
| **Pretests** | | | |
| Pitch threshold_speech (in semitone) | 1.6 (2.5) | 1.8 (2.9) | *p* = .86 |
| Pitch threshold_non-speech (in semitone) | 2.4 (3.6) | 1.9 (3.3) | *p* = .66 |
| Pitch short-term memory span (in number of sounds) | 6.2 (1.4) | 6.7 (1.8) | *p* = .34 |
| MBEA_pitch-based (in %) | 82.2 (9.8) | 81.1 (9.2) | *p* = .75 |
| MBEA_long-term memory (in %) | 91.4 (6.4) | 89.8 (12.4) | *p* = .66 |
| MBEA_overall (in %) | 82.7 (8.9) | 81.6 (9.7) | *p* = .82 |

*Note.* The average scores, with the standard deviation in parenthesis, of the two training groups in each pretest are reported. The results (*p*-value) of independent-samples *t*-tests comparing the two groups in age and scores are also reported.

pretests was conducted for each group to ensure that the morning and evening groups were matched at their pitch-related sensitivity and memory capacity at the group level before the training session. The pretest session consists of three tasks: 1). A pitch threshold test adopted from Ho et al. [64], which was used to measure the participants' low-level pitch sensitivities of speech and non-speech tones; 2). A pitch memory span test adopted from Williamson and Stewart [65], which was used to measure the participants' short-term pitch memory capacities; and 3). The Montreal Battery of Evaluation of Amusia (MBEA) [66], which was used to measure the participants' musical abilities. The MBEA consists of six subtests, three of which are pitch-based tests (scale, contour, and interval), and the remaining three are two duration-based tests (rhythm and meter) and a long-term (musical) memory test. As can be seen in Table 1, the two training groups performed similarly in the pitch threshold task (both speech and non-speech tones), the pitch (short-term) memory span task, the pitch-based MBEA tests, the (long-term) memory MBEA test, and the overall MBEA performance. Results of independent-samples *t*-test confirmed that the scores of the two training groups in each pretest were not significantly different from each other.

The experimental procedures were approved by the Human Subjects Ethics Sub-committee of the PolyU. Informed written consent was obtained from the participants in compliance with the experiment protocols. All the participants were recruited from January to May 2019, and they were paid for their participation.

## 2.2 Stimuli

The stimuli were 30 words contrasting three Cantonese level tones, /55/ T1 (a high-level tone), /33/ T3 (a mid-level tone), and /22/ T6 (a low-level tone). Each tone was carried by ten base syllables (/jan/, /ji/, /jau/, /jiu/, /fan/, /fu/, /ngaa/, /si/, /se/ and /wai/), and all words are meaningful in Cantonese. Each monosyllabic target word was embedded in a carrier phrase context "呢個係_ lei1 go3 hai6 [target word]" (this is [target word]). Two female native Cantonese speakers (see Fig 2) recorded the stimuli: tonal stimuli produced by Talker 1 (i.e., the trained talker) were used in the training session; tonal stimuli produced by both Talker 1 and Talker 2 (i.e., the untrained talker) were used in the assessment posttests. Each speaker recorded three repetitions of each target word in the carrier phrase. Recordings were conducted in a sound-proof room using a microphone linked to a digital recorder. Two tokens for each target word

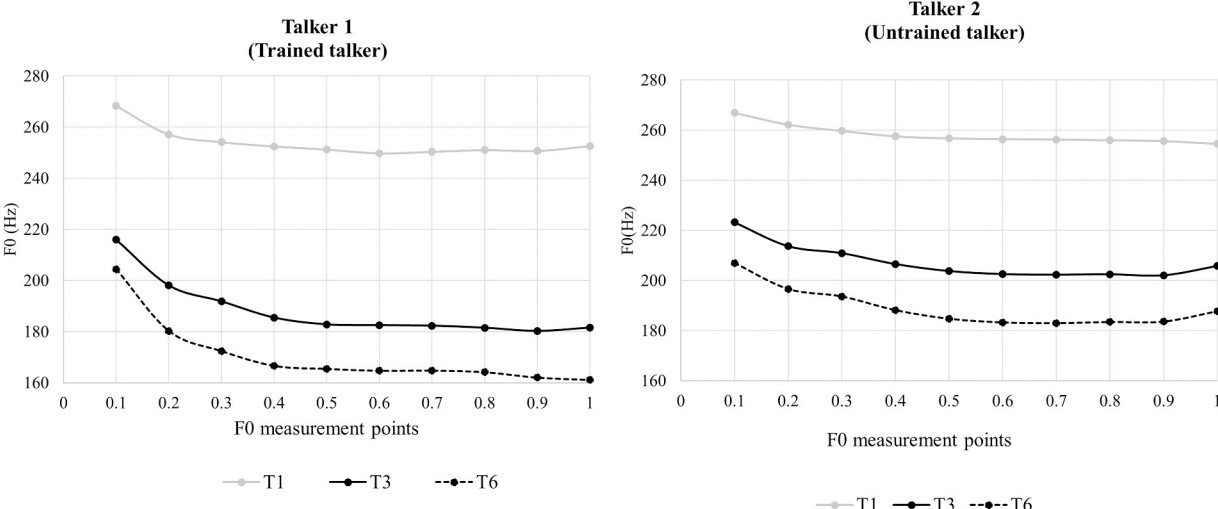

**Fig 2. Tonal contours of the three Cantonese level tones.** Tonal contours were measured using ten measurement points and produced by the trained female talker (left panel) whose stimuli were used in both the training and posttests as well as the untrained female talker (right panel) whose stimuli were used in the posttests alone.

were chosen by the investigators based on its intelligibility and sound quality. Each token was segmented out of the carrier phrase. Then its duration was normalized to 500 ms, a value similar to the duration of natural stimuli and which was used in previous research [48,67], and its mean acoustic intensity was scaled to 70 dB using Praat.

The untrained talker differed phonetically from the trained talker in their voice similarity and pitch distribution, and thus was used as a generalization talker in this study. The trained and untrained talkers were selected from a total of six female talkers used in another project [68]. First, the untrained talker, along with the other four female talkers, was compared against the trained talker in terms of voice similarity by 12 native Cantonese speakers on a scale of 1 to 9 (with 1 meaning very dissimilar and 9 meaning very similar). The untrained talker received the lowest similarity rating score (i.e., 4.70) averaged across raters among the five talkers (rating range: 4.70–7.44 on a scale of 1–9). In addition, as illustrated in Fig 2, the trained and untrained talkers had different pitch distribution with the latter having a higher average pitch (trained: 204 Hz; untrained: 219 Hz) and a narrower pitch range (trained: 170–254 Hz; untrained: 189–258 Hz). Due to the between-talker F0 difference, T6 (low-level tone) produced by the untrained talker had even higher pitch values than T3 (mid-level tone) produced by the trained talker. Therefore, given the voice and pitch distribution dissimilarity between the two talkers, generalizing across the trained and untrained talkers was essential for trainees to learn how to use relative pitch height relationship in categorizing novel level tones after perceptual training.

## 2.3 Procedures

Following the design of previous studies [19,23], a pretest-training-posttest paradigm was conducted on the participants in the two training groups, a morning training group vs. an evening training group. Participants also completed a sleep questionnaire (e.g., sleep duration, sleep quality, fall-asleep time, wake-up time, etc.) for the 24-hour experiment period in the laboratory at the end of the experiment.

The experiments were conducted using the Paradigm software (Perception Research Systems, Inc. http://www.paradigmexperiments.com/). Auditory stimuli were presented to the

participants via stereo headphones with the volume adjusted to a comfortable level. The participants completed the experiments in a quiet space in the Speech and Language Sciences Lab of PolyU.

As mentioned in 2.1, a set of pretests was conducted for each group before training on a separate day (i.e., Day 1). In the training session, a self-paced, forced-choice identification (ID) task of Cantonese level tonal contrasts was conducted on the two training groups. Monosyllables produced by the trained talker (Talker 1 in Fig 2) alone were used in the training session. During the training, the participants were instructed to identify each tone (T1-High, T3-Mid and T6-Low) by pressing three buttons (1, 3, and 6). A total of 300 trials (1 talker * 3 tones * 10 syllables * 2 tokens * 5 repetitions) were auditorily presented to the participants in five blocks with 60 trials in each block. Written feedback ("Correct" in green or "Incorrect. The correct answer is. . ." in red) was given immediately after every trial. The participants were instructed to learn to categorize three tones based on feedback, and achieve the best performance as they can in this session.

In the posttest sessions, the participants were assessed on both ID and AX discrimination. As shown in Table 2, each posttest was conducted at three time points (immediately after training, with a 12-hour delay, and with a 24-hour delay) to test how the learned Cantonese level tones were retained and how the representation of learned tones changed over time after perceptual training. Monosyllables produced by both the trained (Talker 1 in Fig 2) and untrained (Talker 2 in Fig 2) talkers were used in the posttest sessions.

In the ID posttests, the participants were instructed to identify each tone (T1-High, T3-Mid and T6-Low) by pressing three buttons (1, 3, and 6). However, no feedback was given after every trial. The ID posttest consisted of 120 trials (2 talkers * 3 tones * 10 syllables * 2 tokens), which were randomly presented to participants in one block. The ID posttest took approximately 5 min.

Following each ID posttest, participants were tested in an AX discrimination posttest. In the AX discrimination posttests, the participants were instructed to distinguish whether two tones they heard belonged to the same or different tone categories by pressing one of two buttons (left arrow and right arrow) indicating "same" or "different", respectively, on the keyboard. No feedback was given. An equal number of AA pairs (120 pairs with the same tone within each pair) and AB pairs (120 pairs with different tones within each pair) were used to counterbalance the two types of tone pairs. The presentation order of two tones in each AB pair was also counter-balanced in different trials. Following the design of previous research

**Table 2. Overview of timing in the experimental protocol.**

| Days | Day1 | Day 2 | | Day 3 | |
|---|---|---|---|---|---|
| *Time* | | 8-10AM | 8-10PM | 8-10AM | 8-10PM |
| *Sessions* | Pretest session | Session1 | Session2 | Session3 | |
| **Morning Training Group** | 1.Pitch threshold task<br>2.Pitch memory task<br>3. MBEA | AX pretest<br>ID training<br>ID posttest1<br>(immediate)<br>AX posttest1<br>(immediate) | ID posttest2<br>(12-hour delay)<br>AX posttest2<br>(12-hour delay) | ID posttest3<br>(24-hour delay)<br>AX posttest3<br>(24-hour delay) | |
| **Evening Training Group** | 1.Pitch threshold task<br>2.Pitch memory task<br>3. MBEA | | AX pretest<br>ID training<br>ID posttest1<br>(immediate)<br>AX posttest1<br>(immediate) | ID posttest2<br>(12-hour delay)<br>AX posttest2<br>(12-hour delay) | ID posttest3<br>(24-hour delay)<br>AX posttest3<br>(24-hour delay) |

[19], two acoustically different tokens of the same tone type were used in each AA pair such that listeners were required to make tone discrimination on the basis of a change in the membership of tone categories. In addition, a relatively long inter-stimulus interval, 1000 ms, was used for a similar reason to favor the use of phonological knowledge. The design of the AX posttest was intended to tap an individual's recognition of the tone category rather than the use of low-level acoustic information (e.g., pitch) to discriminate tonal tokens. The AX posttest had 240 trials (2 talkers *3 pairs * 2 orders * 10 syllables * 2 types), which were presented in a random order to the participants in one block. The AX discrimination posttest took approximately 20 min.

Following the design of [19], an AX discrimination pretest was conducted as a baseline assessment to make sure that the discrimination ability of the two groups was comparable before the training. During the three posttests, the ID posttest (a shorter duration, e.g., 5 min) was always administered prior to the AX discrimination posttest (a longer duration, e.g., 20 min) to reduce an effect of trainees' fatigue after a long task, with confirmation of feedback from pilot participants, and to maximize the effect of training on both types of posttests. In total, the study involved three sessions of behavioral experiments requiring four lab visits for each participant, which took about four hours in total.

## 2.4 Data analysis

For the ID task, a logit mixed-effects model with Group (2 levels: morning vs. evening; morning as Baseline), Time (3 levels: ID posttest1, ID posttest2, ID posttest3; ID posttest1 as Baseline), Talker (2 levels: trained vs. untrained; trained talker as Baseline), and Tone (3 levels: T1, T3, and T6; T1 as Baseline) as fixed effects, with participants and items as random intercepts and Time as a random slope on the participant random intercept (the models with more random slopes did not converge), was performed on the participants' accuracy data (1 = correct, 0 = incorrect) of their ID posttests.

For the AX discrimination task, *d*-prime scores, which serve to tease apart any response bias from sensitivity and was commonly applied to discrimination of tones [69], were computed to assess the participants' discrimination ability in this study. The *d*-prime score for each participant was derived based on the "hit" rate (number of times the "different" button was pressed for AB pairs) and the "false alarm" rate (number of times the "different" button was pressed for AA pairs). A *d*-prime score is the difference between "hit" rate and the "false alarm" rate when they are *z*-transformed [70]. A linear mixed-effects model with Group (2 levels: morning vs. evening; morning as Baseline), Time (4 levels: pretest, posttest1, posttest2, posttest3; pretest as Baseline), Talker (2 levels: trained vs. untrained; trained talker as Baseline), and Tone Pair (3 levels: T1-T3, T1-T6, and T3-T6; T1-T3 as Baseline) as fixed effects, with participants (item information was obscured by computing *d*-prime scores of the discrimination task, so item was not in the random effect structure) as a random intercept and Time as a random slope on the participant random intercept (the models with more random slopes did not converge), was conducted on the participants' *d*-prime scores.

Note that mixed-effects models (generalized linear mixed models on the identification accuracy and linear mixed models on *d*-prime scores) with a maximal random effects structure used in the current study have greater statistical power than the analysis of variance adopted in previous studies [19,22], and are more appropriate for our data, for instance, binomial data of identification accuracy [71,72]. The models were fitted in R, using the glmer() and lmer() function from the lme4 package for logit and linear mixed-effects models, respectively [71]. A back-fitting function from the package *LMERConvenienceFunctions* in R [73] was used to identify the best model that accounted for significantly more of the variance than simpler

models, as determined by log-likelihood ratio tests; only the results of the model with the best fit are presented, with *p* values calculated using the *lmerTest* package in R [74]. In addition, the observed (post hoc) power to detect each fixed effect of the best model was computed by simulation based on the effect size (*Estimate* column) of this dataset using the *SIMR* package in R (powerSim function with 100 simulations), in the interest of power analyses for future replication and follow-up studies of this research.

Analyses yielding significant interactions between fixed factors (e.g., Group and Time) were followed up by subsequent models conducted separately, for instance, for each group. If there are sleep-mediated changes in the identification and discrimination maintenance (i.e., the overnight consolidation effect), the morning and evening training groups are expected to differ in performance changes over the 24-hour experiment period, yielding a significant two-way interaction between Group and Time. If the potential sleep-mediated changes are only found for stimuli produced by the untrained talker, a significant three-way interaction of Group, Time and Talker is expected. Similar effects in ID and AX discrimination tasks are expected, if different degrees of stimuli variability accounted for the discrepancy of the two tasks in terms of the overnight consolidation effect in the previous study [19]. We predict that the overnight consolidation does not differ across specific level tonal contrasts, so the interaction between Group and Time is not expected to interact with Tone in the ID tasks and Tone Pair in the AX discrimination tasks.

## 3. Results

To ensure that the morning and evening groups had comparable levels of tone discrimination sensitivity before going through training, an independent-samples *t*-test by Group was performed on the accuracy of the AX discrimination pretest, a baseline assessment. The accuracy of the two groups did not significantly differ from each other [*t* (30) = −0.46, *p* = .65].

We do not have a baseline measure of ID performance, because the tone-category pairing would have been random before the participants learned to categorize tones in the training session. Note that the participants in the two groups spent a comparable amount of time in training (Evening: mean = 22.4 min, S.D. = 7.4; Morning: mean = 24.5, S.D. = 4.2), which was not significantly different from each other according to the *t*-test result [*t* (23.6) = −0.99, *p* = .33]. To ensure that the participants showed some learning of the tonal categories following training as intended (i.e., performing above chance), a one-sample *t*-test was conducted on the participants' accuracy in the ID posttest1 immediately after training. The accuracy of the morning group [*t* (15) = 8.80, *p* < .001] and the evening group [*t* (15) = 9.72, *p* = < .001] in the ID Posttest was both statistically above chance (33.3%), indicating that the perceptual training was effective for both training groups. Moreover, the results of an independent-samples *t*-test by Group on the accuracy of the ID posttest1 showed that differences in group performance immediately following training were not statistically significant [*t* (30) = −0.61, *p* = .55], suggesting that an initial learning immediately after training was comparable between the two groups.

### 3.1 ID performance

Fig 3 presents the morning and evening training groups' mean proportions of correct response (i.e., accuracy) for stimuli produced by the trained and untrained talkers in the ID posttests over the 24-hour period (ID posttest1, ID posttest2, and ID posttest3).

Recall that a logit mixed-effects model was performed on participants' accuracy to examine the effects of Group (morning vs. evening), Time (ID posttest1, ID posttest2, ID posttest3), Talker (trained vs. untrained), and Tone (T1, T3, and T6), and their interactions. The baseline

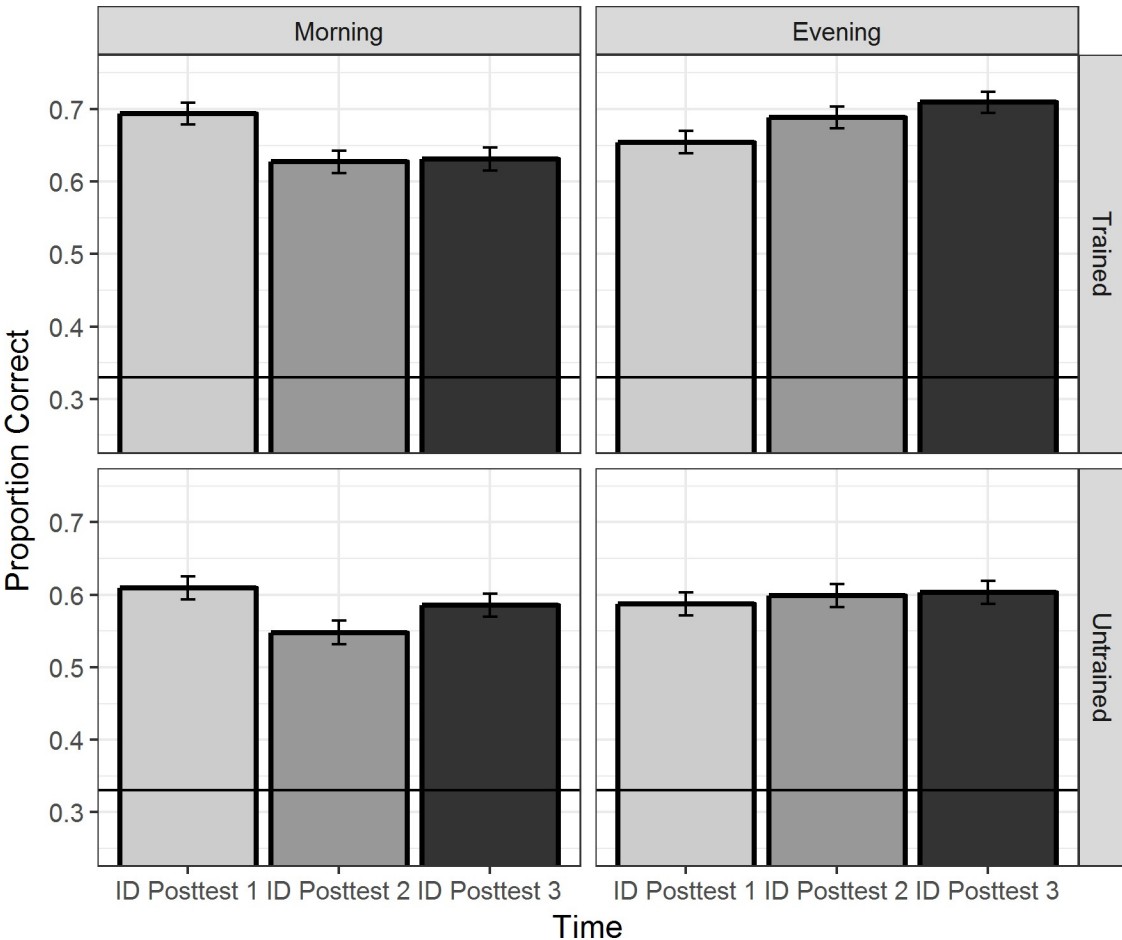

**Fig 3. The morning and evening training groups' mean proportions of correct response for stimuli produced by the trained and untrained talkers over the 24-hour period in the ID posttests.** The error bars represent one standard error of the mean; the horizontal line represents chance performance.

was the morning groups' performance on trained stimuli of T1 in the ID posttest1. The model with the best fit included the simple effects of Group, Time, Tone, Talker, as well as the interaction between Group and Time. The estimate, standard error, $z$ value, and $p$ value associated with the fixed effects are presented in Table 3.

The model results summarized in Table 3 indicate that the morning group's ID performance on trained stimuli of T1 in the ID posttest1 was larger than 0 (intercept); the morning group's performance of accuracy on trained stimuli of T1 in either ID posttest2 or 3 was lower than their performance in the ID posttest1 (simple effect of Time); the morning group's performance in the ID posttest1 on trained stimuli of T3 and T6 was lower than their performance of T1 trials (simple effect of Tone); the morning group's performance on stimuli of T1 in the ID posttest1 for the untrained talker was lower than their performance for the trained talker (simple effect of Talker). Importantly, the model also yielded a significant two-way interaction between Group and Time, indicating that the two groups differed in their ID performance changes over the 24-hour experiment period. However, the three-way interaction among Group, Time and Talker did not improve the model, indicating a lack of evidence that the ID performance changes over time of the two groups differed between stimuli produced by the

**Table 3. Best logit mixed-effects model on proportions of correct response of the participants from the morning and evening groups in the three ID posttests.**

| Effect | Estimate | Standard error | z | p | Observed power |
|---|---|---|---|---|---|
| (intercept) | 1.568 | 0.143 | 10.95 | < .001 | |
| Group | −0.180 | 0.191 | −0.95 | .34 | 16% |
| Time (ID Posttest2) | −0.331 | 0.088 | −3.75 | < .001 | 94% |
| Time (ID Posttest3) | −0.225 | 0.082 | −2.74 | < .01 | 90% |
| Tone | −0.193 | 0.01 | −19.38 | < .001 | 100% |
| Talker | −0.374 | 0.041 | −9.17 | < .001 | 100% |
| Group × Time (ID Posttest2) | 0.444 | 0.125 | 3.562 | < .001 | 94% |
| Group ×Time (ID Posttest3) | 0.387 | 0.115 | 3.359 | < .001 | 90% |

*Note. df* = 11479, 32 participant, 20 items.

trained and untrained talkers. The lack of interaction between Tone and other factors indicates that ID performance changes (e.g., for the morning/evening groups and for trained/untrained stimuli) remained similar for all the level tones.

Since the participants in the morning and evening groups were tested over different time-courses during the 24-hour period, it is not appropriate to compare the performance of each group in each specific test [19,22,23]. Follow-up logit mixed-effects models were therefore performed on the participants' accuracy across three ID posttests separately for each training group to understand the nature of the significant two-way interaction between Group and Time. For the morning group, the participants' performance in either the ID posttest2 (Estimate = –0.320, Std. Error = 0.087, $z$ = –3.68, $p$ < .001, power = 98%) or the ID posttest3 (Estimate = –0.220, Std. Error = 0.072, $z$ = –2.99, $p$ < .01, power = 84%) was significantly lower than their performance in the ID posttest1. However, the participants' performance between the ID posttest2 and the ID posttest3 did not significantly differ (Estimate = 0.106, Std. Error = 0.070, $z$ = 1.51, $p$ = .13, power = 38%). The results of the morning group suggest that participants without an intervening night's sleep between Session 1 (i.e., training session) and Session 2 (i.e., posttest2) showed a significant declining ID performance in the later posttests compared with their ID performance in the initial posttest.

For the evening group, the participants' performance in the ID posttest2 did not differ from their performance in the ID posttest1 (Estimate = 0.109, Std. Error = 0.081, $z$ = 1.33, $p$ = .18, power = 30%). The participants' performance difference between the ID posttest3 and the ID posttest1 had a marginal significance (Estimate = 0.153, Std. Error = 0.082, $z$ = 1.88, $p$ = .06, power = 40%). Again, the participants' performance in the ID posttest2 and the ID posttest3 did not significantly differ (Estimate = 0.043, Std. Error = 0.071, $z$ = 0.61, $p$ = .54, power = 8%). The results of the evening group suggest that participants with an intervening night's sleep between Session 1 (i.e., training session) and later sessions showed a numerical trend of improved ID performance in posttest2, and a marginally significant improvement in posttest3, compared with their ID performance in the initial posttest. However, there is no evidence that their ID performance improved during the 12 waking hours between posttest2 and posttest3. Importantly, the different perceptual patterns over time between the morning and evening training groups were found for the stimuli produced by both the trained and untrained talkers.

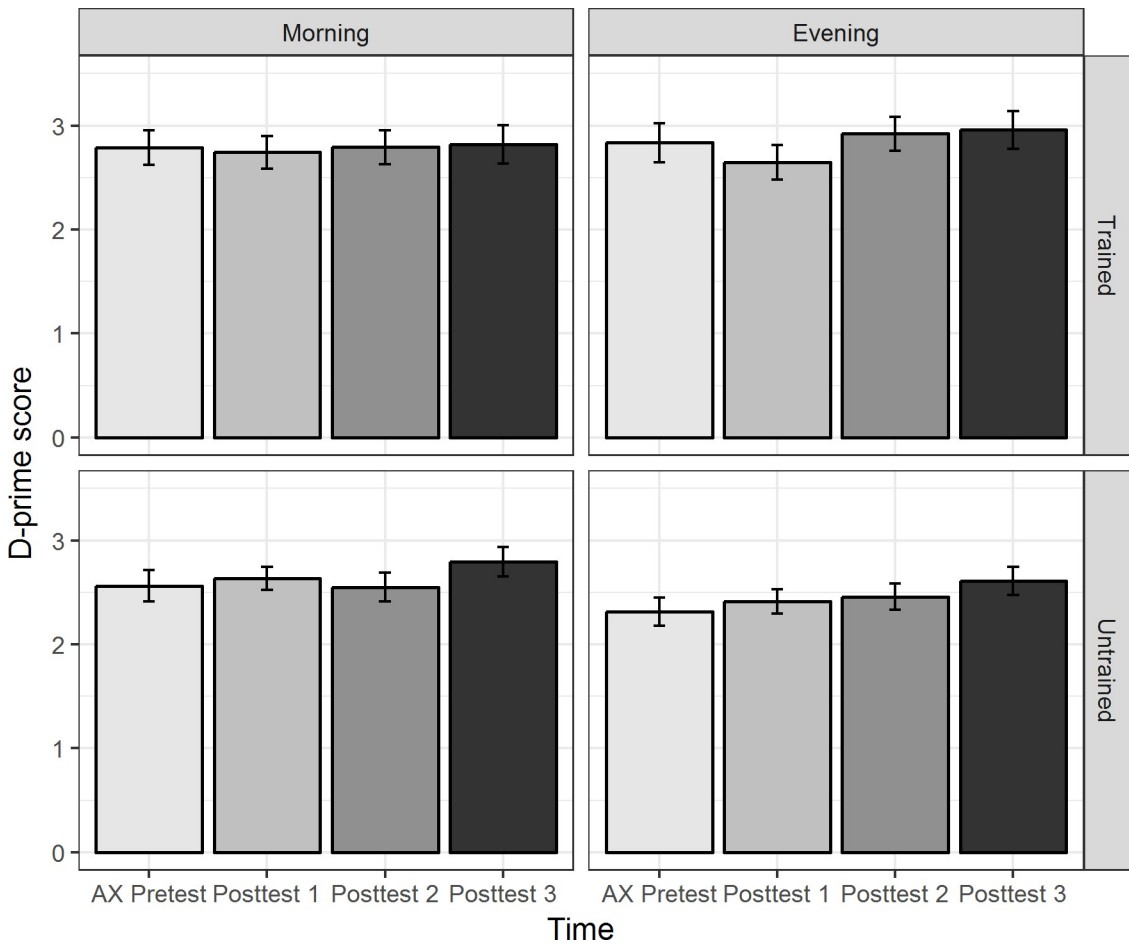

**Fig 4. The morning and evening training groups' mean _d_-prime scores for stimuli produced by the trained and untrained talkers over the 24-hour period in the AX discrimination pretest and posttests.** The error bars represent one standard error of the mean.

### 3.2 AX discrimination performance

Fig 4 presents the morning and evening training groups' mean _d_-prime scores for stimuli produced by the trained and untrained talkers in the AX discrimination tests over the 24-hour period (AX pretest, AX posttest1, AX posttest2, and AX posttest3).

Recall that a linear mixed-effects model was performed on participants' _d_-prime scores to examine the effects of Group (morning vs. evening), Time (AX pretest, AX posttest1, AX posttest2, and AX posttest3), Talker (trained vs. untrained), and Tone Pair (T1-T3, T1-T6, and T3-T6), and their interactions. The baseline was the morning groups' performance on trained stimuli of T1-T3 in the AX discrimination pretest. The model with the best fit included the simple effects of Group, Tone Pair, Talker, and the interaction between Group and Talker as well as between Tone Pair and Talker. The estimate, standard error, _t_ value, and _p_ value associated with the fixed effects are presented in Table 4.

The model results summarized in Table 4 indicate that the morning group's performance on trained stimuli of T1-T3 pairs in the AX discrimination pretest was larger than 0 (intercept); the morning group's performance in the pretest on trained stimuli of T1-T6 pairs or T3-T6 pairs was higher than their performance of T1-T3 pairs (simple effects of Tone Pair);

the morning group's performance on the untrained stimuli of T1-T3 pairs in the pretest was lower than their performance for the trained stimuli of T1-T3 pairs (simple effect of Talker).

The model also yielded significant two-way interaction effects of Group and Talker as well as of Tone Pair (T3-T6) and Talker. Crucially, the interaction between Group and Time did not improve the model, indicating that the two groups did not differ in AX discrimination performance changes over the 24-hour experiment period. The three-way interaction among Group, Time and Talker did not improve the model neither, indicating that the two groups showed similar discrimination performance changes over time for stimuli produced by both the trained and untrained talkers.

To understand the nature of the significant two-way interaction between Group and Talker, subsequent linear mixed-effects models were therefore performed on the participants' $d$-prime scores for stimuli produced by the trained and untrained talkers separately for each training group. As illustrated in Fig 4, for the morning group, the participants' performance for the trained and untrained talkers did not significantly differ (Estimate = −0.150, Std. Error = 0.101, $t$ = −1.48, $p$ = .14, power = 28%). For the evening group, the participants' performance for the trained talker was significantly higher than their performance for the untrained talker (Estimate = −0.392, Std. Error = 0.098, $t$ = −3.99, $p$ < .001, power = 100%). The results suggest that the evening group had a greater difficulty discriminating the stimuli produced by the untrained talker, who had a narrower pitch range as well as smaller pitch differences between some tones (e.g., T1 and T3) than the trained talker (see Fig 2). Importantly, the perceptual pattern was the same across the pretest and posttests without Time modulating it. Thus, the finding was not attributed to an overnight consolidation effect in facilitating talker-independent tone learning.

To understand the nature of the significant two-way interaction between Tone Pair (T3-T6) and Talker, subsequent linear mixed-effects models were therefore performed on the participants' $d$-prime scores for stimuli produced by the trained and untrained talkers separately for T1-T3 and T3-T6. As illustrated in Fig 5, for T1-T3 pairs, the participants' performance for stimuli produced by the untrained talker was significantly lower than their performance for stimuli produced by the trained talker (Estimate = −0.530, Std. Error = 0.061, $t$ = −8.66, $p$ < .001, power = 100%). For T3-T6 pairs, the participants' performance for stimuli produced by the trained and untrained talker did not significantly differ (Estimate = 0.104, Std. Error = 0.056, $t$ = 1.87, $p$ = .06, power = 50%). The results suggest that all the participants found T1-T3 pairs produced by the untrained talker, which had smaller pitch differences than T1-T3 pairs produced by the trained talker (see Fig 2), perceptually more difficult than the

**Table 4. Best linear mixed-effects model on $d$-prime scores of the participants from the morning and evening groups in the pretest and three posttests of AX discrimination.**

| Effect | Estimate | Standard error | df | t | p | Observed power |
|---|---|---|---|---|---|---|
| (intercept) | 3.245 | 0.121 | 44.86 | 26.76 | < .001 | |
| Group | 0.056 | 0.162 | 36.06 | 0.34 | .73 | 8% |
| Tone Pair (T1-T6) | 0.351 | 0.068 | 736 | 5.18 | < .001 | 100% |
| Tone Pair (T3-T6) | −1.729 | 0.068 | 736 | −25.53 | < .001 | 100% |
| Talker | −0.408 | 0.078 | 736 | −5.22 | < .001 | 100% |
| Group × Talker | −0.243 | 0.078 | 736 | −3.10 | < .01 | 80% |
| Tone Pair × Talker (T1-T6) | 0.143 | 0.096 | 736 | 1.49 | .14 | 28% |
| Tone Pair × Talker (T3-T6) | 0.634 | 0.096 | 736 | 6.62 | < .001 | 100% |

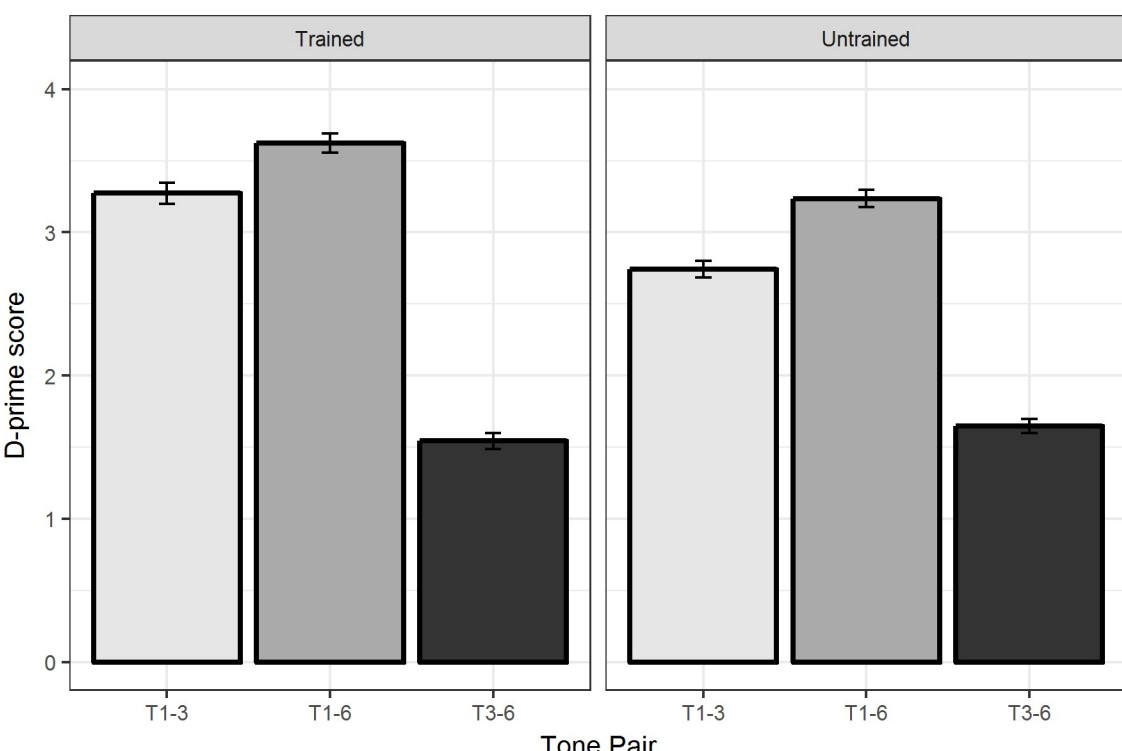

**Fig 5. The participants' mean *d*-prime scores of T1-T3, T1-T6, and T3-T6 trials produced by the trained talker (left panel) and the untrained talker (right panel).** The error bars represent one standard error of the mean.

counterparts of the trained talker. Importantly, the perceptual pattern was the same across the training groups and the pretest-posttests without Group (groups with vs. without intervening night's sleep between training and posttest2) and Time modulating it. Thus, the finding was not attributed to an overnight consolidation effect.

### 3.3 The effect of sleep-related variables

This analysis focused on the evening group to examine the relation of their performance changes over the 24-hour period to their sleep. The results of the sleep questionnaire showed that the participants in the evening group had on average 7.3-hour of sleep (range: 6–8 hours). They got an average of 7.6 (range: 6–9) as a self-rating score of their sleep quality on a 1–10 point scale (with 1 meaning very bad and 10 meaning very good). The participants self-reported to fall asleep at 00:40 am on average (range: 11:30 pm to 2:30 am), and to wake up at 7:50 am on average (range: 7:00 am -8:30 am).

A set of regression analyses were carried out to examine to what extent the performance changes of the participants in the evening group can be predicted by their sleep-related variables. Because sleep-related variables are highly collinear, self-reported sleep duration, sleep quality rating, sleep time (what time did the participants fall asleep) and wake-up time (what time did the participants wake up) were entered stepwise as predictors into a multiple regression model using SPSS software with performance changes (the difference between posttest2 and posttest1; the difference between posttest3 and posttest1) as the dependent variables for each task.

The results of regression analyses showed, as illustrated in Fig 6, that sleep time accounted for the largest proportion of variance in overnight identification changes, that is, the difference

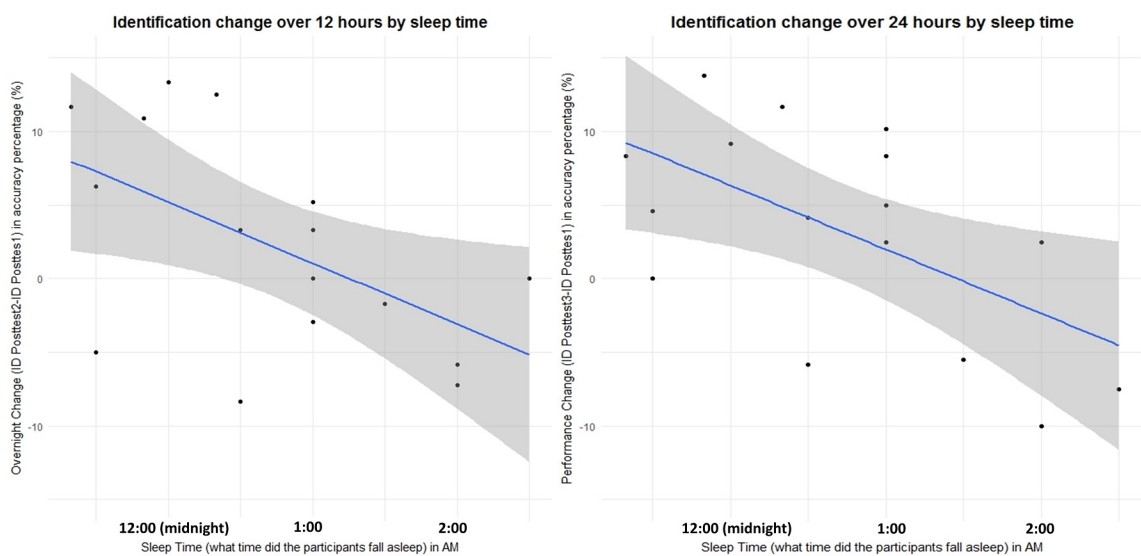

**Fig 6. Relationships between individual differences in sleep time (in AM) and the performance (accuracy percentage in %) changes over 12 hours (left) and 24 hours (right) to the identification task performance.** The shaded areas represent a 5% confidence interval.

between posttest2 and posttest1 [F (1.14) = 5.991, $p$ = .028, $r^2$ = .537]. Sleep time also accounted for the largest proportion of variance in the identification changes between posttest3 and posttest1 [F (1.14) = 6.992, $p$ = .019, $r^2$ = .577]. No sleep-related variables significantly accounted for the variance in discrimination changes. The findings suggest that the earlier the participants fell asleep, the more identification gains they would have in the ID posttests on the following day.

## 4. Discussion and conclusion

The current study examined whether overnight consolidation facilitates talker-independent tone learning, implying an abstraction of novel tonal information, in the identification and discrimination of novel Cantonese level-level tonal contrasts for Mandarin listeners. The results of identification tasks showed that the Mandarin listeners who were trained in the morning, without an intervening night's sleep between training and Posttest2, showed a declining performance of identification accuracy; in contrast, the Mandarin listeners who were trained in the evening, with an intervening night's sleep between training and later posttests, showed a numerical trend of improved performance in identification accuracy, predicted by their individual sleep time. Crucially, the pattern was found for stimuli produced by both the trained and untrained talker across all three level tones. On the other hand, the results of discrimination tasks did not reveal divergent performance changes for the two training groups.

First and foremost, the findings suggest that the sleep intervening training and posttests facilitated talker-independent tone learning. That is, the difference of post-training identification changes between the evening and morning trainees was not only found for stimuli produced by the trained talker, but also for stimuli produced by the novel untrained talker. This finding implies that sleep-mediated consolidation might have assisted the evening trainees' generalization to stimuli produced by the untrained talker. This finding of talker-independent learning seems to support a sleep-mediated consolidation process from acoustic-phonetic (i.e., pitch) features at an episodic level to a formation of a more abstract representation of novel

tone categories in long-term (e.g., declarative) memory. This possibility is in line with the previous findings that overnight consolidation promoted listeners' speech sound abstraction in perceptual learning and lexically-guided phonetic retuning of non-native segments [15,19,23].

More specifically, compared with the morning trainees, the sleep protected the evening trainees' identification of novel tonal contrasts against subsequent decay, and further induced a trend of sleep-related improved identification performance. The finding suggests that sleep-mediated consolidation might have exerted either a protective or restorative effect on the evening trainees' post-training performance of tone identification. The protective effect found in perceptual learning of novel tones is consistent with previous findings in phonetic learning [11,12].

In addition, the trend of improved identification performance over time found for the evening trainees is also (partially) in line with previous research on perceptual learning of segments which showed an improved identification performance after sleep [19]. More importantly, the evening trainees' magnitude of improvement in post-training identification performance was predicted by their individual sleep time. The finding suggests that the earlier the trainees fell asleep, the more likely the post-training identification performance improved in the perceptual learning of Cantonese level tones. The significant relationship between the identification performance changes of the evening trainees and their sleep time provided further evidence supporting the facilitation account of sleep-mediated consolidation. This finding is also consistent with that of recent studies showing that sleep-related variables (e.g., total sleep duration) predicted behavioral and neural changes of adult speech learning of non-native segments after a perceptual training in the evening [20]. While an earlier sleep time is possibly correlated with a higher sleep quality and/or a longer sleep duration, self-rated sleep quality scores and self-reported sleep duration were not found to predict the identification performance changes of the evening trainees in the current study. It is possible that the questionnaire method adopted in the current study renders the sleep duration and quality data less precise, and an objective tool (e.g., a sleep-monitoring headband) is thus suggested to be used to further investigate whether, and if so how, different sleep-related variables affect the evening trainees' performance changes over time.

Note that although there was a trend of improved identification performance over the 24-hour period for the evening trainees, it was not statistically significant. The discrepancy between this finding and that of previous studies could be accounted by the difference of target stimuli. One possible explanation is that the variable and dynamic nature of lexical tones may have made them more difficult to learn than segments. As a result, different from previous studies (on perceptual learning of dental and retroflex stops) which found that the morning trainees did not change their identification performance over time [19], the morning trainees in the current study exhibited declining post-training identification performance over time. In a similar vein, this account may explain why the evening trainees exhibited a non-significant trend of improvement in their post-training identification performance, while managing to maintain the training-induced tone learning after sleep. Given the potential difficulty of tone learning, future studies may consider recruiting a larger sample of evening trainees and using a longer training session to test whether improved post-training tone identification will be found at a group level after sleep-mediated consolidation.

Different from the results of the previous study [19] which found the different perceptual changes of the two training groups in identifying the stop stimuli produced by the untrained talker alone, we found divergent performance changes of the two groups in identifying the level-tone stimuli produced by both the trained and untrained talkers. This discrepancy can be potentially attributed to the different nature of segments (e.g., stops) and lexical tones. As mentioned earlier, given the variable and dynamic nature of lexical tones, tones might be more

difficult for participants to learn/abstract than segments. As a result, compared with the morning groups who showed a declining identification performance, the intervening sleep between training and later posttests may have helped the evening trainees maintain their learning of the stimuli produced by the trained talker, which were also perceptually challenging [25,26,29,30]. An alternative (but related) explanation is that the training stimuli in the current study had a greater variability than those in the previous study [19]. While this study used two tokens of each tone carried by ten different syllables as the training stimuli, the previous study [19] used five tokens of each stop carried by a single syllable (i.e., vowel context). In addition, three-way tonal contrasts were trained as target stimuli in the current study whereas a two-way segmental contrast was used in the previous study. These differences might have made the novel tonal categories perceptually more challenging to learn/abstract, and thus contribute at least partially to the discrepancy regarding the effect for the trained talker's stimuli [12]. Previous tone training studies found that stimulus variability of syllables and speaker would modulate the perceptual learning outcome [54,75–78]. Thus, a future direction of this line of research is to investigate whether a greater or a smaller acoustic variability (of syllables and/or speakers) of training stimuli would modulate the size of the overnight consolidation effect in facilitating the abstraction of novel phonetic information.

An important question that the current results raise is why the two training groups showed divergent performance changes over time in the identification tasks, but not in the discrimination tasks, which were designed to tap into listeners' phonological knowledge as well by using two different tokens of the same tone category in AA pairs and a relatively long ISI. First, previous studies on segments claimed that the lack of group differences in the discrimination tasks was probably due to its greater stimuli variability (i.e., untrained vowel contexts) than those in the identification tasks [19,22]. Nonetheless, the findings of the current study did not seem to support this account as identical tonal stimuli were used in both the identification and discrimination tasks. It is possible that the overnight consolidation effect in terms of abstraction is more related to the identification (i.e., explicit recall of tone category labels through declarative memory) than the discrimination (i.e., implicit ability to detect acoustic cue differences in the signal) of novel sound categories in nature [15].

Another plausible explanation is the different nature of training (identification) and assessment (discrimination) tests, which tapped into different aspects of non-native tone perception. The identification task used in the training session (and posttests) tapped into higher levels of phonological encoding of tonal categories. In contrast, the discrimination tasks used in the posttests, albeit intended to test an individual's phonological processing of tone categories, might still have tapped into relatively low levels of phonetic processing instead. Previous training research found that the phonetic processing in sound discrimination did not change much even after multiple perceptual training sessions [79–81]. Related to this plausibility, the divergent performance changes found in the identification tasks, but not in the discrimination tasks, might be due to the nature of the identification training. Previous studies showed that a categorization (or identification) training of non-speech stimuli resulted in an "acquired similarity", that is, a decrease of sensitivity to within-category differences; in contrast, an alternative discrimination training of the same set of stimuli yielded an increase of sensitivity to within-category differences [82]. Similar findings regarding the identification versus discrimination training methods were also reported for training of non-native tones [83]. In the current study, the identification training may have assisted the detection of shared features/dimensions of stimuli in the same tone category and a sleep-mediated consolidation process then facilitated this abstraction of tone categories. Therefore, it is not surprising that the two training groups showed divergent performance changes in the identification task associated with an overnight consolidation effect. Instead, the discrimination task involved a detection of

acoustic differences between tones, and it is reasonable that the two training groups had comparable performance in this task, for which both groups were not trained. Future studies may consider employing both an identification task as well as a discrimination task with feedback in the training session to further examine whether an effect of overnight consolidation is restricted to high-level phonological abstraction (e.g., identification) in nature or can be ultimately yielded in the low-level sensitivity assessments (e.g., discrimination).

Lastly, note that a talker effect was found for the two training groups in the AX discrimination tasks but not in the ID tasks, which requires an explanation. Compared with the morning trainees, who exhibited comparable discrimination performance for the trained and untrained talkers, the evening trainees' discrimination accuracy for the trained talker was significantly higher than the untrained talker. The evening trainees' performance may be attributed to their less optimal arousal level or reduced attentional resources for perception when tested in the evening [15,22], especially for perceptually challenging and less familiar stimuli since the untrained talker had a narrower pitch range as well as smaller pitch differences between some tones (e.g., T1 and T3). Intriguingly, this talker effect was not found in the ID task, presumably because the ID task was shorter in duration and less demanding in working memory (note that the discrimination of two stimuli requires the temporary storage of auditory stimuli in working memory).

To conclude, to the best of our knowledge, the present study is the first to examine the effect of overnight consolidation in perceptual learning of lexical tones. The present findings suggest that sleep-mediated overnight consolidation might facilitate the talker-independent learning of Cantonese level tones by promoting the tone learning from the trained talker to the untrained talker. Sleep protected evening trainees' identification from subsequent decay, and yielded identification performance changes, which were correlated with their individual sleep time, over the 24-hour experiment period. These findings raise several questions regarding the training stimuli, sleep-related variables and others for further research. Among other things, it would be intriguing to investigate the potential effect of overnight consolidation in perceptual learning of different types of tone pairs (e.g., contour-contour tone pairs vs. level-level tone pairs) and other prosodic categories (e.g., lexical stress) to further shed light on the effect of overnight consolidation in perceptual learning of suprasegmental domain as well as different nature of segments and prosodic categories in this process.

## Acknowledgments

This work was supported in part by the Departmental General Research Funds (International collaboration) and the Departmental Reward Scheme for Research Publications in Indexed Journals awarded to CCZ, and the Language Learning Early Career Research Grant and the Postdoctoral Fellowships Scheme at the Department of Chinese and Bilingual Studies of the Hong Kong Polytechnic University awarded to ZQ.

## Author Contributions

**Conceptualization:** Zhen Qin, Caicai Zhang.

**Data curation:** Zhen Qin.

**Formal analysis:** Zhen Qin.

**Funding acquisition:** Caicai Zhang.

**Investigation:** Zhen Qin, Caicai Zhang.

**Methodology:** Zhen Qin.

**Writing – original draft:** Zhen Qin.

**Writing – review & editing:** Caicai Zhang.

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
