## [Decision Letter · Decision Letter 0]

18 Sep 2019

PONE-D-19-22122

The effect of overnight consolidation in the perceptual learning of non-native tonal contrasts

PLOS ONE

Dear Dr. Zhang,

Thank you for submitting your manuscript to PLOS ONE. After careful consideration, we feel that it has merit but does not fully meet PLOS ONE’s publication criteria as it currently stands. Therefore, we invite you to submit a revised version of the manuscript that addresses the points raised during the review process.

Both reviewers have expressed that the research presented can have valuable impact, but have suggestions for improving the strength of the paper and interpretations. Both reviewers offer suggestions for improving the clarity of the paper, particularly with regards to providing justification for certain methodological decisions. In particular, Reviewer 2 also recommends incorporating more coverage of the background literature on the effects of sleep on learning to improve the theoretical justification behind your predictions, and has questions and ideas about the interpretation of your results. I agree that the suggestions brought forth by the reviewers would improve the article, and thus offer the opportunity to revise the manuscript in accordance with their feedback.

We would appreciate receiving your revised manuscript by Nov 02 2019 11:59PM. To enhance the reproducibility of your results, we recommend that if applicable you deposit your laboratory protocols in protocols.io, where a protocol can be assigned its own identifier (DOI) such that it can be cited independently in the future. For instructions see: http://journals.plos.org/plosone/s/submission-guidelines#loc-laboratory-protocols

We look forward to receiving your revised manuscript.

Kind regards,

Karen E. Mulak, Ph.D.

Academic Editor

PLOS ONE

Journal Requirements:

Reviewers' comments:

Reviewer's Responses to Questions

**Comments to the Author**

1. Is the manuscript technically sound, and do the data support the conclusions?

Reviewer #1: Partly

Reviewer #2: Partly

2. Has the statistical analysis been performed appropriately and rigorously? 

Reviewer #1: Yes

Reviewer #2: I Don't Know

3. Have the authors made all data underlying the findings in their manuscript fully available?

Reviewer #1: Yes

Reviewer #2: Yes

4. Is the manuscript presented in an intelligible fashion and written in standard English?

Reviewer #1: Yes

Reviewer #2: Yes

5. Review Comments to the Author

Reviewer #1: Following previous work on the effects of sleep vs. wake state on the consolidation of learned segmental nonnative speech contrasts, the authors examined the potential effect of sleep on the generalization of tones across talkers. To this end, the authors trained two groups of learners: Evening and Morning, and examined group-level differences in the pattern of changes to perceptual performance over 24 hours. The authors found that while those trained in the Evening appeared to improve in their identification performance over the subsequent 24 hours, that the Morning group declined in performance. The authors interpret their findings as evidence of sleep’s role in talker generalization.

I found this work to be interesting and well-executed. I have some methodological questions, and comments/minor suggestions for edits.

Methods:

Why was raw trial-by-trial accuracy modeled for the ID task, whereas AX performance appears to have first been transformed into d’? I can see pros/cons of each (and thus have no objection to either being used, per se) – but I do find the inconsistency in how the data was handled across tasks to be a bit odd.

Which sleep questionnaire was used (in-house? If so, what were the questions asked?)

Interpretation:

The authors rightly point out some important distinctions between the current findings and the Earle & Myers (2015) study on talker generalization. Specifically, the current findings of a Time*Group interaction (but no Time*Group*Talker interaction) indicates that performance on the untrained talker follows a roughly similar learning trajectory to the trained talker, if I’m understanding correctly. Given that, talker generalization of tones may not be separate from talker-specific learning of tones. Given that, some scaling back of the assertion that the findings are primarily about talker generalization might be warranted (e.g. “First and foremost…overnight consolidation facilitates generalization across talkers…”(p.25)). The findings that sleep promotes perceptual tone learning in general is in itself interesting.

I do agree with the authors’ interpretation that some differences with previous findings may relate to the variability of stimuli used during training (p.28). The findings by Fenn et al. (2013) cited elsewhere in the manuscript may be of some relevance to this point.

Minor comments:

Table 3 in the ID performance section says “AX posttests” – which results are being displayed here?

The authors use the term “development” to refer to changes to perceptual behavior across time (in abstract and in the introduction). As “developmental” is most often used to refer to changes during child development, the use of the term here comes across as confusing.

Figures (at least on my copy) appear to have low-resolution/are blurry.

Reviewer #2: Summary

This study explored the effects of memory consolidation after sleep on the perceptual learning of new lexical tone contrasts. Two groups of native Mandarin speakers were trained to identify 3 Cantonese level tones [T1 (high-level), T3 (mid-level) and T6 (low-level)]. One group was trained in the morning and tested immediately after training, after a 12-hour delay and after a 24-hour delay. The other group was trained in the evening and tested immediately after training, 12 hours later after a night of sleep, and again in the evening, after a 24-hour delay. Both groups were administered pitch sensitivity and pitch memory tasks and an AX perceptual discrimination task before the identification (ID) training. After training, they were administered both the AX as well as the ID posttests. The results showed a significant increase in tone identification accuracy from posttest 1 to posttest 3 for both trained and untrained talker among the evening training group but not among the morning training group. No significant difference was found between the two groups in the AX discrimination posttests. The authors concluded that “overnight consolidation facilitates generalization across talkers in the identification of novel Cantonese level-level tonal contrasts.”

Comments

The effects of memory consolidation resulting from sleep on learning among animals and humans remains controversial. While extending this line of research to lexical tone learning will add new information to an existing body of research in speech learning, concerns on some aspects of the research design and the interpretations of the results prevent me from recommending this paper for publication in its current forms. But with substantial revision, the authors may be able to contribute with this data set.

In case it is helpful in future revisions, some of my primary concerns are outlined below:

1. The introduction did not provide enough information to motivate the theoretical predictions. I suggest that the authors thoroughly review the literature on memory consolidation during sleep for information on such issues as the type of memory (i.e., declarative versus episodic) facilitated by sleep, and how this may impact speech sound category learning. Mechanisms underlying an identification (ID) and an AX discrimination task should also be thoroughly discussed. It’s not clear to me what the theoretical motivation was for including both tasks in the study. Different levels (e.g., auditory-acoustic, phonetic and phonological) of processing should also be discussed. From the literature, an ISI of 1 sec would likely elicit the phonetic or the phonological level of processing (as opposed to the auditory level of processing), making the AX and the ID tasks comparable. Yet, the different findings between the two tasks are puzzling. In addition, it’s also not clear why acoustic dynamicity, claimed to be characteristics of lexical tones but not of segments (vowel and consonants), would differentially impact the overnight consolidation effects. Note that vowel and consonant acoustic properties are also dynamic and are subject to variations across phonetic contexts, talkers, speaking rates, etc.

2. Some of the results are not consistent with the general conclusion that “overnight consolidation facilitates generalization across talkers in the identification of novel Cantonese level-level tonal contrasts” while some others remain to be accounted for. As pointed out by Fenn et al. (2003), “Sleep has at least two separate effects on learning. Sleep consolidates memories, protecting them against subsequent interference or decay. Sleep also appears to ‘recover’ or restore memories “ (p. 616). From Figure 3, for trained speaker, the difference in identification accuracy between posttest1 and posttest2 for the evening training group was not statistically significant, suggesting, contrary to the authors’ conclusion, a lack of consolidation during the 12-hour period of sleep. Significant improvement from posttest1 to posttest3 is puzzling in so far as it suggested that additional consolidation occurred during the 12 waking hours between posttest2 and posttest3. Note also that the morning training group outperformed the evening training group numerically immediately after training (postest1). In fact, even in posttest2, the evening training group’s performance remained numerically lower than that of the morning group at posttest1. It would appear that there is a ceiling effect of the training. While the evening training group still has room to improve after posttest1, the morning training group appears to have hit a ceiling.

3. Statistical power is also a main concern. The authors should present evidence of enough statistical power of all the statistical analyses performed.

4. It is also peculiar why the talker asserted different effects on the two training groups and on the two types of tests (ID and AX discrimination).

5. A number of sentences are not easily parsed so the paper should be proof-read by a native speaker.

Minor questions/concerns

1. Did the subjects in each training group spend the same amount of time on the training?

2. Why was it necessary to make sure that the subjects performed at an above chance level immediately after training?

3. Why were the tone stimuli normalized to 500 ms?

4. Discuss if and how the administration of the ID posttests prior to the AX discrimination posttest may have been responsible for the results obtained.

6. PLOS authors have the option to publish the peer review history of their article (what does this mean?). If published, this will include your full peer review and any attached files.

Reviewer #1: No

Reviewer #2: No

---

## [Author Response · Author response to Decision Letter 0]

28 Oct 2019

Dear Dr. Mulak,

We would like to thank you for giving us the opportunity to revise and resubmit our manuscript, and the reviewers for their insightful comments on our manuscript. Below, we elaborate on how we have addressed the reviewers’ concerns. We also refer to specific changes of the marked-up copy (e.g., page number), highlighted in red, wherever appropriate. 

We believe the manuscript is now stronger thanks to the reviewers’ comments. We hope that you will find the revised manuscript satisfactory, and look forward to hearing back from you soon.

Sincerely,

Zhen Qin, Caicai Zhang

Re: comments of the Associate Editor

Both reviewers have expressed that the research presented can have valuable impact, but have suggestions for improving the strength of the paper and interpretations. Both reviewers offer suggestions for improving the clarity of the paper, particularly with regards to providing justification for certain methodological decisions. In particular, Reviewer 2 also recommends incorporating more coverage of the background literature on the effects of sleep on learning to improve the theoretical justification behind your predictions, and has questions and ideas about the interpretation of your results. I agree that the suggestions brought forth by the reviewers would improve the article, and thus offer the opportunity to revise the manuscript in accordance with their feedback.

• We followed both reviewers’ suggestions by providing justifications for our methodological decisions (e.g., Reviewer 1’s question about dependent variables; Reviewer 2’s concerns regarding the two tasks used in posttest sessions). 

• In responses to Reviewer 2’s concerns about the theoretical justification behind our predictions, more background literature on the effects of sleep tested in novel word learning and speech sound learning have been added in the introduction section to give a more comprehensive review on the facilitating role of memory consolidation during sleep. 

• We followed Reviewer 2’s suggestion, by either adding additional interpretation of our results (e.g., sleep-mediated consolidation protects evening trainees against subsequent decay), or revising our interpretation to clarify it if we opted to keep (a part of) our original interpretation (i.e., the findings of generalization to untrained talker as well as the evening trainees’ sleep-related identification changes altogether support the effect of sleep-mediated consolidation). 

Re: comments of Reviewer 1

Methods:

Why was raw trial-by-trial accuracy modeled for the ID task, whereas AX performance appears to have first been transformed into d’? I can see pros/cons of each (and thus have no objection to either being used, per se) – but I do find the inconsistency in how the data was handled across tasks to be a bit odd.

• We decided to stick to our analyses on the accuracy modeled for the ID task and d-prime scores used for the AX discrimination task after checking the relevant literature and doing additional analyses on the accuracy of the discrimination task. 

• In response to Reviewer 1’s concern regarding dependent variables, a sentence justifying our use of d-prime scores for the AX discrimination task was added (p.19) in the data analysis section. 

• Specifically, a d-prime score serves to tease apart any response bias from sensitivity, and was commonly applied to discrimination of tones in previous studies (Francis & Ciocca, 2003). Moreover, models on the trial-by-trial accuracy of the AX task yielded results (e.g., no effect of overnight consolidation effect) similar to models on the d-prime scores of the AX tasks. Therefore, we decided to stick to our original analyses on the d-prime scores used for the AX discrimination task.

Which sleep questionnaire was used (in-house? If so, what were the questions asked?)

• No, participants completed a sleep questionnaire for the 24-hour experiment period in the laboratory, instead of in-house, at the end of the experiment. We now clarified this information and questions asked in the procedures section (p.15). 

• A sample of our sleep questionnaire, together with our data, was uploaded in open science website (https://osf.io/284j9) for the further interest of reviewers and readers. 

Interpretation:

The authors rightly point out some important distinctions between the current findings and the Earle & Myers (2015) study on talker generalization. Specifically, the current findings of a Time*Group interaction (but no Time*Group*Talker interaction) indicates that performance on the untrained talker follows a roughly similar learning trajectory to the trained talker, if I’m understanding correctly. Given that, talker generalization of tones may not be separate from talker-specific learning of tones. Given that, some scaling back of the assertion that the findings are primarily about talker generalization might be warranted (e.g. “First and foremost…overnight consolidation facilitates generalization across talkers…”(p.25)). The findings that sleep promotes perceptual tone learning in general is in itself interesting.

• We thank the reviewer for suggesting some scaling back of the assertion regarding “talker generalization”. We have replaced “talker generalization” with “talker-independent tone learning” throughout the manuscript and specified what we mean by “talker generalization” in terms of the overnight consolidation effect. 

I do agree with the authors’ interpretation that some differences with previous findings may relate to the variability of stimuli used during training (p.28). The findings by Fenn et al. (2013) cited elsewhere in the manuscript may be of some relevance to this point.

• Thanks for this suggestion. We have added this reference to support the explanation of the variability of stimuli used during training.

Minor comments:

Table 3 in the ID performance section says “AX posttests” – which results are being displayed here?

• This has been fixed. 

The authors use the term “development” to refer to changes to perceptual behavior across time (in abstract and in the introduction). As “developmental” is most often used to refer to changes during child development, the use of the term here comes across as confusing.

• This has been fixed. As suggested, “development” was revised as “post-training perceptual changes” so that confusion would not arise.

Figures (at least on my copy) appear to have low-resolution/are blurry.

• This has been fixed. 

 

Re: comments of Reviewer 2

1. The introduction did not provide enough information to motivate the theoretical predictions. I suggest that the authors thoroughly review the literature on memory consolidation during sleep for information on such issues as the type of memory (i.e., declarative versus episodic) facilitated by sleep, and how this may impact speech sound category learning. 

• Thanks to the reviewer for raising the points. Following the reviewer’s suggestion, more studies on novel word learning (e.g, Tamminen, Davis, Merkx, & Rastle, 2012) and speech sound learning (Fenn, Margoliash, & Nusbaum, 2013; e.g., Fenn, Nusbaum, & Margoliash, 2003) were reviewed in terms of the impact of sleep on the type of memory (i.e., the abstraction from episodic information to context-independent representation in long-term memory through an overnight consociation effect) before going into the details of Earle’s studies in the introduction part. 

• In addition, a two-stage complementary learning systems (CLS) model (McClelland, McNaughton, & O’Reilly, 1995) was explicitly discussed to clarify how sleep might impact language (both words and speech sounds) learning in a general context of the sleep-mediated memory consolidation.

• To make the introduction more theoretically focused, experimental details of some studies (e.g., Xie, Earle, & Myers, 2018) were also condensed if the study was not directly related to the current study.

…Mechanisms underlying an identification (ID) and an AX discrimination task should also be thoroughly discussed. It’s not clear to me what the theoretical motivation was for including both tasks in the study. 

• In response to Reviewer 2’s comments about the ID and AX discrimination tasks used in the posttests, we have more fully motivated the two tasks in the section of literature review (Earle & Myers, 2015a, 2015b) and research questions

• First, to clarify the theoretical motivation for including both tasks in previous studies, specifically, we have added the text (p. 6) to read “the identification task, relying on explicit recall of sound categories, and the category discrimination task, using a listener’s implicit ability to automatically direct attention toward the acoustic cues in the signal, were argued to reflect different aspects of listeners’ memory system, and thus were both included in posttest sessions (Earle & Myers, 2014, 2015a).” Since we did not fully agree with the hypothesis of a dichotomy of declarative memory vs. procedural memory tested by the identification and discrimination tasks, and it is not the focus of our current study, this dichotomy in detail was not mentioned in the revised manuscript to avoid any potential confusion.

• Second, to motivate the use of the two tasks in our study, we have revised our motivation (p.10-11) to read “As discussed above, the magnitude of the overnight consolidation effect and underlying mechanisms may differ between the identification and discrimination tasks (Earle & Myers, 2014, 2015a). For this reason, we included both identification and discrimination tasks to test whether the overnight consolidation effects found for the identification can be transferred or generalized to the discrimination of tonal contrasts by controlling stimuli variability in both tasks.”

Different levels (e.g., auditory-acoustic, phonetic and phonological) of processing should also be discussed. From the literature, an ISI of 1 sec would likely elicit the phonetic or the phonological level of processing (as opposed to the auditory level of processing), making the AX and the ID tasks comparable. Yet, the different findings between the two tasks are puzzling. 

• We agree with the reviewer’s comments, that is, “an ISI of 1 sec would likely elicit the phonetic or the phonological level of processing (as opposed to the auditory level of processing), making the AX and the ID tasks comparable.” We now clarified our original intention in making the AX discrimination tasks comparable to the ID tasks in the procedure section (p.18). That is, following the design used in previous research (Earle & Myers, 2015a), a category discrimination with a relatively long ISI (1000 ms) was used in the current study intending to favor the use of phonologic knowledge like the identification task. 

• Meanwhile, following the reviewer’s suggestions, we now emphasized the possible account in terms of different levels (e.g., auditory-acoustic and phonological) of processing in the discussion (p.38-39) to account for the different findings between the two tasks. We acknowledged that the results of the two tasks (i.e., the presence vs. lack of overnight consolidation effect and the talker effect) showed that the two talks may have actually tapped into listeners’ higher-level phonological knowledge (in the identification task) and low-level acoustic sensitivity (in the discrimination task) respectively. 

….In addition, it’s also not clear why acoustic dynamicity, claimed to be characteristics of lexical tones but not of segments (vowel and consonants), would differentially impact the overnight consolidation effects. Note that vowel and consonant acoustic properties are also dynamic and are subject to variations across phonetic contexts, talkers, speaking rates, etc.

• To clarify, we did not intend to make general claims that lexical tones are dynamic and subject to variations across phonetic contexts, talkers, speaking rates, etc., whereas segments are not. Instead, lexical tones are relatively more dynamic and variable than segments in this context, specified as the Hindi dental and retroflex stops, which were tested in previous research on the overnight consolidation effect (Earle & Myers, 2015a, 2015b). The text (p.8) has been revised to clarify this information.

• Moreover, references comparing tones to segments were added (p.8) in the introduction section to help justify our focus on lexical tones in examining the overnight consolidation effect. For instance, according to previous research (e.g, Singh, Goh, & Wewalaarachchi, 2015; Wewalaarachchi, Wong, & Singh, 2017; Wiener & Turnbull, 2016), the variable and dynamic features of lexical tones were reported to induce a higher degree of sensitivity to mispronunciations of tones (i.e., a change in tone identity) than those of vowels and consonants in spoken word recognition for children and adult listeners. 

2. Some of the results are not consistent with the general conclusion that “overnight consolidation facilitates generalization across talkers in the identification of novel Cantonese level-level tonal contrasts” while some others remain to be accounted for. 

As pointed out by Fenn et al. (2003), “Sleep has at least two separate effects on learning. Sleep consolidates memories, protecting them against subsequent interference or decay. Sleep also appears to ‘recover’ or restore memories “ (p. 616). From Figure 3, for trained speaker, the difference in identification accuracy between posttest1 and posttest2 for the evening training group was not statistically significant, suggesting, contrary to the authors’ conclusion, a lack of consolidation during the 12-hour period of sleep. 

• We agree with Reviewer 2 that there might be two separate effects on learning. As suggested, the learning effect in terms of protecting the evening group against subsequent decay which was experienced by the morning group, was added in the discussion (p.33) and conclusion (p.40). 

• More importantly, to clarify our main argument, our conclusion of the sleep-mediated consolidation effect is drawn based on the primary finding of the generalization to the untrained talker as well as the evening trainees’ sleep-related identification changes, rather than whether the evening trainees improved their post-training identification. The discussion section was re-organized to highlight our primary finding for talker-independent learning (p.32-33), that is, “the difference of post-training identification changes between the evening and morning trainees was not only found for stimuli produced by the trained talker, but also for stimuli produced by the novel untrained talker.” 

• To clarify our second argument supporting the sleep-mediated consolidation effect, the evening trainees’ sleep-related identification changes were discussed in relation to the regression results of the magnitude of changes and sleep time (p.33-34). 

• In response to the reviewer’s concern regarding the lack of significant improvement in identification between posttest1 and posttest2, the discussion of the evening trainees’ trend of identification changes was scaled down to reflect our current results (i.e., a numerical trend of improvement). We also acknowledged the discrepancy between our results and previous studies in terms of the significantly improved performance in the discussion, and proposed a possible explanation for it (i.e., the variable and dynamic nature of lexical tones may have made them more difficult to learn than segments) (p. 34-35). 

Significant improvement from posttest1 to posttest3 is puzzling in so far as it suggested that additional consolidation occurred during the 12 waking hours between posttest2 and posttest3. 

• Thanks to the reviewer for pointing out this possibility. As shown in the revised analyses (see the next bullet point for details), there was a numerical trend of improvement between posttest2 and 1 as well as between posttest3 and 1.

• In the interest of consistency with the large mixed-effects models, all post-hoc pairwise comparisons are now conducted with a maximal random structure (i.e., with Time as a random slope on the participant random intercept). With the revised analyses, the results of post-hoc analyses come out slightly different: now while the difference in identification accuracy between posttest1 and posttest2 for the evening training group was not statistically significant (z = 1.33; p =.18), the difference between posttest1 and posttest3 was marginally significant (z = 1.88; p =.06). Like in the original analyses, no significant difference was found between posttest2 and posttest3 (z = 0.61; p =.54) for the evening group.

• Note that there was no significant difference between posttest2 and posttest3. These results provided evidence against the possibility of additional consolidation during the waking period. A sentence was added (p.26) to clarify this information. 

Note also that the morning training group outperformed the evening training group numerically immediately after training (postest1). In fact, even in posttest2, the evening training group’s performance remained numerically lower than that of the morning group at posttest1. It would appear that there is a ceiling effect of the training. While the evening training group still has room to improve after posttest1, the morning training group appears to have hit a ceiling.

• We agree with the reviewer’s observation of the numerical difference between the morning and evening groups in posttest1 as well as in posttest2. This observation was consistent with previous research (Earle & Myers, 2015; Xie et al., 2017), which also showed a numerical advantage of the morning group over the evening group. The morning group’ advantage is something that generally cannot be avoided since the two groups were tested at different time points. 

• To clarify, since the participants in the morning and evening groups were tested across different time-courses during the 24-hour period, it is not appropriate to compare the performance of each group in each specific test given their different physiological (e.g., arousal level) states. Therefore, consistent with previous research, the present study focused on comparing the changes of performance across different tests for each training group rather than comparing the two groups in each test. A sentence was added (p.25) in the revised manuscript to justify our suggested analyses. 

• The results of pretests and training have ensured that the morning and evening groups had comparable levels of tone discrimination sensitivity before training, and a similar degree of improvement immediately after training. Lastly, the ID accuracy of the morning group (Mean: 0.66; SD: 0.4) and their AX d-prime scores (Mean: 2.68; SD: 1.1) in posttest1 suggest that the morning group did not have a ceiling performance. Thus, we are confident that the results shall not be attributed to the morning training group having a ceiling performance at the beginning while the evening group had more room for improvement. 

3. Statistical power is also a main concern. The authors should present evidence of enough statistical power of all the statistical analyses performed.

• In responses to Reviewer 2’s concerns regarding statistical power, the (post hoc) observed power to detect each fixed effect of the models was computed by simulation based on the effect size (‘Estimate’ column of Tables) of this dataset using R-package simr (powerSim function with 100 simulations), and then reported in the manuscript. As can be seen in the results, most of the significant effects we found had a high power (e.g., above 80%). 

• To clarify our use of the observed power, it is difficult for us to use the effect size reported in previous studies to calculate the power of each effect in our analyses. Different statistical analyses (e.g., ANOVA) were used in previous studies, and the effect size was not fully reported in all analyses of previous studies (Earle & Myers, 2015; Xie et al., 2017). The observed power, together with effect size, was used for the following aspects: 1). Estimating true power of each significant effect found in our study; 2). concluding that there is no effect when it did not reveal a statistical difference; 3). facilitating power analyses for future replication and follow-up studies of this research. The justification was also added (p.20-21) in the revised manuscript. 

• In addition, we felt that our sample size and statistical analyses might also justify enough statistical power compared with previous studies (Earle & Myers, 2015b, 2015a). The justification of sample size was added (p.12) to read as follows: “Although no power analysis was performed for calculation of sample size, the sample size of the morning and evening participants in the current study was largely comparable to that reported in studies on previous studies (Earle & Myers, 2015; Xie et al., 2017).” 

• The justification of statistical analyses was added (p.20) to read as follows: “Note that mixed-effects models (generalized linear mixed models on the identification accuracy and linear mixed models on d-prime scores) with a maximal random effects structure used in the current study have greater statistical power than the analysis of variance adopted in previous studies (Earle & Myers, 2015c, 2015a), and are more appropriate for our data, for instance, binomial data of identification accuracy (Baayen, 2008; Linck & Cunnings, 2015). ”

4. It is also peculiar (strange) why the talker asserted different effects on the two training groups and on the two types of tests (ID and AX discrimination).

• To address the talker effect found for the AX discrimination rather than in the ID task, a possible explanation was added (p.39-40), in the discussion of the revised manuscript, to read as follows: “compared with the morning trainees, who exhibited comparable discrimination performance for the trained and untrained talkers, the evening trainees’ discrimination accuracy for the trained talker was significantly higher than the untrained talker. The evening trainees’ performance may be attributed to their less optimal arousal level or reduced attentional resources for perception when tested in the evening (Earle & Myers, 2014, 2015c), especially for perceptually challenging and less familiar stimuli since the untrained talker had a narrower pitch range as well as smaller pitch differences between some tones (e.g., T1 and T3). Intriguingly, this talker effect was not found in the ID task, presumably because the ID task was shorter in duration and less demanding in working memory (note that the discrimination of two stimuli requires the temporary storage of stimuli in working memory).”

5. A number of sentences are not easily parsed so the paper should be proof-read by a native speaker.

• This has been fixed.

Minor questions/concerns

1. Did the subjects in each training group spend the same amount of time on the training?

• Yes, The participants in each training group spent a comparable amount of time (Evening: mean=22.4 min, S.D.=7.4; Morning: mean=24.5, S.D.=4.2). An independent t-test showed that the two groups’ training duration was not significantly different [t (23.6) =-0.99, p = .33]. This information was added (p.22) in the revised manuscript.

2. Why was it necessary to make sure that the subjects performed at an above chance level immediately after training?

• The results that the participants performed above chance suggested that 

they showed some learning of the tonal categories following training as intended. It is also suggestive evidence that the training session was equally effective for both groups. We have revised the text (p.22) to specify the purpose.

3. Why were the tone stimuli normalized to 500 ms?

• 500 ms was chosen as a value of duration normalization in previous research on Cantonese tone perception (Zhang, Shao, & Huang, 2017), and was reported to be similar to the duration of natural Cantonese stimuli (Mok, Zuo, & Wong, 2013). Thus, it was chosen for duration normalization. We now justified our choice in the stimuli section (p.14).

4. Discuss if and how the administration of the ID posttests prior to the AX discrimination posttest may have been responsible for the results obtained.

• Thanks to the reviewer for raising the point. First, we provided justification of the task order in the procedure section (p.18). Specifically, the order of the ID and AX posttests was decided based on the following considerations: 1). the order of the ID posttests prior to the AX discrimination posttest was consistent with previous research testing the effect of overnight consolidation (Earle & Myers, 2015a); 2). the participants tested in a pilot study reported that they felt less tired to have an ID posttest (5 min) administered first, instead of an AX discrimination posttest (20 min) first. To maximize the effect of perceptual training, we felt that it would be optimal to administer the ID posttests prior to the AX discrimination posttests. 

• Regarding the possible influence of the task order, given a significantly longer duration of the AX discrimination task than the ID task, it is reasonable to believe that the ID task would have exerted much less influence on the AX discrimination than the other way around. Our results showed that while perceptual changes were found across the ID tasks, the pattern was not found across the AX discrimination tasks. We take this result to support our view that it is not case that the results of the AX discrimination were driven by the ID results. In the interest of space, we did not discuss possible results of an alternative order in the revised manuscript as justification of the task order was already provided. 

  

References

Baayen, R. H. (2008). Analyzing Linguistic Data: A Practical Introduction to Statistics Using R. Harald Baayen (2008). Sociolinguistic Studies. https://doi.org/10.1558/sols.v2i3.471

Earle, F. S., & Myers, E. B. (2014). Building phonetic categories: An argument for the role of sleep. Frontiers in Psychology, Vol. 5, pp. 1–12. https://doi.org/10.3389/fpsyg.2014.01192

Earle, F. S., & Myers, E. B. (2015a). Overnight consolidation promotes generalization across talkers in the identification of nonnative speech sounds. The Journal of the Acoustical Society of America, 137(1), EL91–EL97. https://doi.org/10.1121/1.4903918

Earle, F. S., & Myers, E. B. (2015b). Sleep and native language interference affect non-native speech sound learning. Journal of Experimental Psychology: Human Perception and Performance. https://doi.org/10.1037/xhp0000113

Earle, F. S., & Myers, E. B. (2015c). Sleep and native language interference affect non-native speech sound learning. Journal of Experimental Psychology: Human Perception and Performance, 41(6), 1680–1695. https://doi.org/10.1037/xhp0000113

Fenn, K. M., Margoliash, D., & Nusbaum, H. C. (2013). Sleep restores loss of generalized but not rote learning of synthetic speech. Cognition, 128(3), 280–286. https://doi.org/10.1016/j.cognition.2013.04.007

Fenn, K. M., Nusbaum, H. C., & Margoliash, D. (2003). Consolidation during sleep of perceptual learning of spoken language. Nature, 425(6958), 614–616. https://doi.org/10.1038/nature01951

Francis, A. L., & Ciocca, V. (2003). Stimulus presentation order and the perception of lexical tones in Cantonese. The Journal of the Acoustical Society of America. https://doi.org/10.1121/1.1603231

Linck, J. A., & Cunnings, I. (2015). The Utility and Application of Mixed-Effects Models in Second Language Research. Language Learning. https://doi.org/10.1111/lang.12117

McClelland, J. L., McNaughton, B. L., & O’Reilly, R. C. (1995). Why there are complementary learning systems in the hippocampus and neocortex: Insights from the successes and failures of connectionist models of learning and memory. Psychological Review, 102(3), 419–457. https://doi.org/10.1037/0033-295X.102.3.419

Mok, P. P. K., Zuo, D., & Wong, P. W. Y. (2013). Production and perception of a sound change in progress: Tone merging in Hong Kong Cantonese. Language Variation and Change, 25(3), 341–370. https://doi.org/10.1017/s0954394513000161

Singh, L., Goh, H. H., & Wewalaarachchi, T. D. (2015). Spoken word recognition in early childhood: Comparative effects of vowel, consonant and lexical tone variation. Cognition, 142, 1–11. https://doi.org/10.1016/j.cognition.2015.05.010

Tamminen, J., Davis, M. H., Merkx, M., & Rastle, K. (2012). The role of memory consolidation in generalisation of new linguistic information. Cognition, 125(1), 107–112. https://doi.org/10.1016/j.cognition.2012.06.014

Wewalaarachchi, T. D., Wong, L. H., & Singh, L. (2017). Vowels, consonants, and lexical tones: Sensitivity to phonological variation in monolingual Mandarin and bilingual English–Mandarin toddlers. Journal of Experimental Child Psychology, 159, 16–33. https://doi.org/10.1016/j.jecp.2017.01.009

Wiener, S., & Turnbull, R. (2016). Constraints of Tones, Vowels and Consonants on Lexical Selection in Mandarin Chinese. Language and Speech, 59(1), 59–82. https://doi.org/10.1177/0023830915578000

Xie, X., Earle, F. S., & Myers, E. B. (2018). Sleep facilitates generalisation of accent adaptation to a new talker. Language, Cognition and Neuroscience. https://doi.org/10.1080/23273798.2017.1369551

Zhang, C., Shao, J., & Huang, X. (2017). Deficits of congenital amusia beyond pitch: Evidence from impaired categorical perception of vowels in Cantonese-speaking congenital amusics. PLoS ONE, 12(8), 1–24. https://doi.org/10.1371/journal.pone.0183151

---

## [Decision Letter · Decision Letter 1]

21 Nov 2019

PONE-D-19-22122R1

The effect of overnight consolidation in the perceptual learning of non-native tonal contrasts

PLOS ONE

Dear Dr. Zhang,

Thank you for submitting your manuscript to PLOS ONE. After careful consideration, we feel that it has merit but does not fully meet PLOS ONE’s publication criteria as it currently stands. Therefore, we invite you to submit a revised version of the manuscript that addresses the points raised during the review process.

The manuscript was sent out for review to Reviewer 2. I agree with Reviewer 2 in saying that the paper has been very much improved and is nearly ready for publication. Reviewer 2 requests that you add discussion of an alternative explanation from findings not involving speech stimuli for your finding of a group difference in the identification task and not the discrimination task, based on the nature of the training and different task requirements. I agree that this argument would fit and complement the paper, and will accept the paper pending this minor revision.

We would appreciate receiving your revised manuscript by Jan 05 2020 11:59PM. To enhance the reproducibility of your results, we recommend that if applicable you deposit your laboratory protocols in protocols.io, where a protocol can be assigned its own identifier (DOI) such that it can be cited independently in the future. For instructions see: http://journals.plos.org/plosone/s/submission-guidelines#loc-laboratory-protocols

We look forward to receiving your revised manuscript.

Kind regards,

Karen E. Mulak, Ph.D.

Academic Editor

PLOS ONE

Reviewers' comments:

Reviewer's Responses to Questions

**Comments to the Author**

1. If the authors have adequately addressed your comments raised in a previous round of review and you feel that this manuscript is now acceptable for publication, you may indicate that here to bypass the “Comments to the Author” section, enter your conflict of interest statement in the “Confidential to Editor” section, and submit your "Accept" recommendation.

Reviewer #2: (No Response)

2. Is the manuscript technically sound, and do the data support the conclusions?

Reviewer #2: (No Response)

3. Has the statistical analysis been performed appropriately and rigorously? 

Reviewer #2: (No Response)

4. Have the authors made all data underlying the findings in their manuscript fully available?

Reviewer #2: (No Response)

5. Is the manuscript presented in an intelligible fashion and written in standard English?

Reviewer #2: (No Response)

6. Review Comments to the Author

Reviewer #2: (No Response)

7. PLOS authors have the option to publish the peer review history of their article (what does this mean?). If published, this will include your full peer review and any attached files.

Reviewer #2: Yes: Ratree Wayland

---

## [Author Response · Author response to Decision Letter 1]

22 Nov 2019

1. One final suggestion I would like to make is for the authors to entertain another possible explanation for the “divergent performance changes over time in the identification tasks, but not in the discrimination tasks” between the two training groups. It has been shown (e.g., Guenther et al., 1999) that, for non-speech stimuli, a categorization training “in which subjects learn to identify stimuli within a particular frequency range as members of the same category, can lead to a decrease in sensitivity to stimuli in that category”, whereas a discrimination training using the same set of stimuli lead to “an increase in sensitivity to differences” among the training stimuli. In other words, “acquired similarity” is observed in the identification training, while the opposite effect is observed in the discrimination training. Since the identification training promotes detection of features/dimensions that the two stimuli shared, then it would make sense for the two groups to differ on the identification task due to the overnight consolidation effect, but they would be comparable in a task that promotes detection of acoustic differences for which they were not trained. This account is similar, but not identical to the ‘explicit recall of tonal category’ vs ‘implicit ability to detect acoustic differences’ explanation that the authors have already proposed.

• We followed Reviewer 2’s suggestion, and added another possible explanation accounting for the “divergent performance changes over time in the identification tasks, but not in the discrimination tasks” between the two training groups (p. 32-33).

• Specifically, as suggested, the different underlying processes of the identification and discrimination training involving non-speech stimuli were discussed. References (Wayland & Li, 2008) on the effect of the training methods on non-native tones were also added. In addition, the effect of our identification training on the two training groups’ performance in the tone identification and discrimination task was elaborated in terms of the training/task nature.

---

## [Editor Report · Decision Letter 2]

26 Nov 2019

The effect of overnight consolidation in the perceptual learning of non-native tonal contrasts

PONE-D-19-22122R2

Dear Dr. Zhang,

We are pleased to inform you that your manuscript has been judged scientifically suitable for publication and will be formally accepted for publication once it complies with all outstanding technical requirements.

With kind regards,

Karen E. Mulak, Ph.D.

Academic Editor

PLOS ONE
---

## [Editor Report · Acceptance letter]

4 Dec 2019

PONE-D-19-22122R2 

The effect of overnight consolidation in the perceptual learning of non-native tonal contrasts 

Dear Dr. Zhang:

I am pleased to inform you that your manuscript has been deemed suitable for publication in PLOS ONE. Congratulations! Your manuscript is now with our production department. 

With kind regards,

on behalf of

Dr. Karen E. Mulak 

Academic Editor

PLOS ONE